# Morphometry and Debris-Flow Susceptibility Map in Mountain Drainage Basins of the Vallo di Diano, Southern Italy

**Salvatore Ivo Giano \*** , **Eva Pescatore and Vincenzo Siervo**

Dipartimento di Scienze, University of Basilicata, via Ateneo Lucano 10, I-85100 Potenza, Italy; eva.pescatore@alice.it (E.P.); vsiervo@gmail.com (V.S.)
* Correspondence: ivo.giano@unibas.it

**Abstract:** In watershed mountain basins, affected in the last decades by strong rainfall events, the role of debris-flow and debris flood processes was investigated. Morphometric parameters have proven to be useful first-approximation indicators in discriminating those processes, especially in large areas of investigation. Computation of morphometric parameters in 19 watershed mountain basins of the western side valley of the Vallo di Diano intermontane basin (southern Italy) was carried out. This procedure was integrated by a semi-automatic elaboration of the potential susceptibility to debris flows, using Flow-R modelling. This software, providing an empirical model of the preliminary susceptibility assessment at a regional scale, was applied in many countries of the world. The implementation of Flow-R modelling requires a GIS application and some thematic base maps extracted using DEMs analysis. A 5-meter-resolution DEM has been used in order to produce the susceptibility maps of the whole study area, and the results are compared and discussed with the real debris flow/flood events that occurred in 1993, 2005, 2010, and 2017 in the studied area. The results have provided a good reliability of Flow-R modelling within small catchment mountain basins.

**Keywords:** debris flows; morphometric parameters; Flow-R modelling; susceptibility maps; regional scale; Vallo di Diano basin; southern Italy

## 1. Introduction

The morphological evolution of drainage basins in mountain areas is mainly controlled by fluvial processes and valley slope movements, such as rockfall and debris flow, representing the most important processes in the valley slope evolution. Among different slope processes, the debris flows are the most dangerous due to their fast movement and long-runout distribution [1]. Debris flow was originally defined as a mass movement of moisture containing granular solids and relatively small amounts of admixed water and/or air ([1–5] and references therein). The shape of debris flows varies strongly in places and, besides their spatial variability, their temporal pattern also appears to be greatly variable. The magnitude–frequency relationships of debris flows make them one of the most dangerous processes occurring on slope valleys [1–5]. As a consequence, debris flows became relevant in hazards when producing damage to buildings, infrastructures and human lives. For this reason, and in the first instance, it is necessary to identify the debris flow susceptibility of a given area based on the availability of field and numerical data [5]. Many studies were devoted to the modelling of debris flow susceptibility at a regional and medium scale analysis, ranging from 1:10,000 to 1:50,000 map scales [6,7], thus providing preliminary analysis of potentially unstable areas and the down-slope regions probably affected by flows [1]. Regional case studies, based on empirical or semi-empirical parameters [8–10], allowed many authors to calibrate the model into small areas where the inventory of past events existed, thus applying the better parameters set ([11] and references therein). In this sense, the use of Flow-R software—a GIS-based process for assessing the regional susceptibility of debris flow—for the identification of potential source areas and the corresponding propagation extent in a medium-scale drainage basin, represents an

interesting method in the preliminary analysis for debris flow susceptibility evaluation, considering the minimum data required [11,12]. The Flow-R, developed in Windows and Linux systems, is a freeware software available at www.flow-r.org (accessed on 27 December 2020) and is based on the Matlab® (Natick, MA, USA) stand-alone application, thanks to the Matlab Compiler Runtime [11]. It was used in different countries and by several universities and national geological services to realize regional maps of debris flow susceptibility with good results [12–19].

The evaluation of morphometric parameters, such as linear, relief and aerial aspects, is a useful mean in the understanding of long- to short-term drainage basin development. Moreover, the results of geomorphic analyses are strictly dependent on geological and geomorphological features of the area. In many cases, the computation of morphometric parameters requires previous field surveys and a photo-aerial interpretation of drainage basins [20–23]. In this sense, the morphometric analysis describes and compares the physical properties of catchment basins and their processes, thus, explaining their geomorphological evolution [19]. Morphometric parameters, as identified by several authors, are reliable predictors for differentiating between debris flow-related and no debris flow-related depositional processes in fan landforms and drainage basin areas, overall [20–23]. Considering that fluvial landforms are the result of drainage basin processes under the action of water over time, self-evident relationships exist between morphometric parameters and fluvial discharge. An evaluation of relationships is very useful in areas where no direct field measurements are available and can be used in the debris flow susceptibility parametrization. The study is aimed at the elaboration of potential susceptibility maps at medium–small scale mountain drainage watersheds located in the western side of the Vallo di Diano basin, in southern Italy (Figure 1). A 5-m-resolution digital elevation model (DEM) elaborated with a GIS-based approach was used in the maps' elaboration. The analysis was based, in the first time, on the computation of the morphometric parameters useful in the recognition of debris flow susceptibility areas. In the second time, it focuses on the application of the Flow-R model, allowing the regional susceptibility assessment of debris flow to be obtained through the identification of potential source areas and corresponding propagation extent [11,15]. The two different approaches are then discussed in order to verify the reliability of the methods.

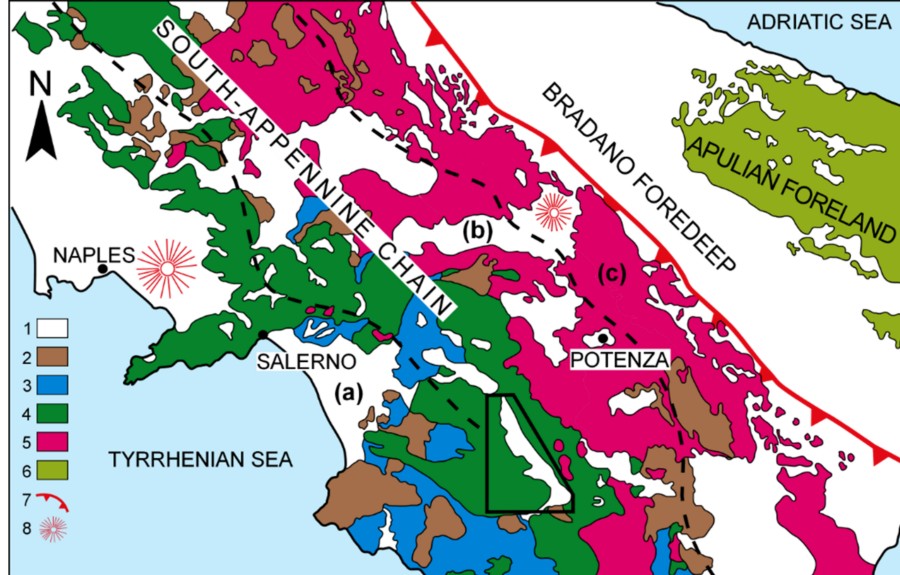

**Figure 1.** Geological sketch map of Southern Apennines and location of the study area. Legend: (1) Plio-Quaternary clastic and volcanic deposits, (2) Miocene syntectonic deposits, (3) Cretaceous to

Oligocene ophiolite-bearing internal units, (4) Meso-Cenozoic shallow-water carbonates of the Apennine platform, (5) Lower–Middle Triassic to Miocene shallow-water and deep-sea successions of the Lagonegro units, (6) Meso-Cenozoic shallow-water carbonates of the Apulian platform, (7) thrust front of the chain, (8) volcanoes. Dark dashed lines indicate boundaries among the inner (**a**), axial (**b**), and outer (**c**) portions of the belt. Dark line box bounds the studied area.

## 2. Materials and Methods

### 2.1. Morphometric Parameters

The morphometric analysis of mountain drainage basins is one of the main approaches in areas potentially affected by mass movement processes. The interpretation of morphometric parameters is a useful way in discriminating debris flow vs. debris flood processes where scarce information on the depositional pattern of debris successions are available and this procedure is widely applied in many areas of the world [20–28]. Debris flows are gravity-induced mass movements that are intermediate between landslides and stream flows and with mechanical characteristics different from either of these processes [8]. According to Costa (1988) [8], debris flows are non-Newtonian visco-plastic or dilatant fluids with laminar flow and uniform concentration profiles, sediment concentrations ranging from 47% to 70% by volume and shear strengths greater than about 40 N/m$^2$. Debris floods are Newtonian fluids with turbulent flow, non-uniform sediment concentration profiles, sediment concentrations less than about 20% by volume and shear strengths less than 100 dines/cm$^2$ [22]. The list of morphometric parameters measured in both the fan depositional areas and the contributing basins were computed in Table 1, as follows:

**Table 1.** Morphometric parameters computed in the western side of the Vallo di Diano basin.

| Basin Area | Ba | Planimetric Area of the Basin Measured above the Fan Apex |
|---|---|---|
| Maximum elevation | Hmax | maximum elevation point measured at the crest of the basin |
| Minimum elevation<br>Mean elevation | Hmin<br>Hmean | minimum elevation point of the basin, measured up the fan apex<br>mean elevation point measured in the basin |
| Basin relief | Bf | vertical difference between the maximum and minimum elevation of the basin |
| Watershed length | Wl | length of the planimetric straight-line measured from the fan apex to the most distant point on the watershed boundary |
| Melton's ruggedness number | R | index of the basin expressed by the following algorithm:<br>$R = H_b A_b^{-0.5}$<br>where *Hb* is the basin relief and *Ab* is the planimetric area of the basin |
| Fan toe | Ft | elevation a.s.l. in meters of the fan toe |
| Fan gradient | Fg | average gradient measured along the longitudinal fan axes |
| Fan area | Fa | planimetric area of the fan measured below the fan apex |

Morphometric parameters have been directly extracted from a 5-m-resolution DEM of the Campania Region and further, they have been locally controlled on 1:25,000 contour maps of the Istituto Geografico Militare Italiano (I.G.M.I.). Moreover, geomorphic evidence of debris flow/flood activity in some selected fans of the piedmont area have been investigated in the field to detect depositional lobes a few meters thick, isolated boulders rolling on the fan surface, massive gravels and boulders filling and overrunning the main roads. More in general, the sedimentary architecture of clastic deposits, which are diagnostics for depositional activity, has been carried out.

With the aim to define a fast way of discriminating which catchment basins could be potentially affected by debris flow or fluvial flood processes, a simple prompt assessment was carried out, plotting our data within three already existing diagrams proposed by different authors. De Scally and Owens (2004) [24] distinguished debris flow from fluvial fans processes in the Southern Alps of New Zealand, plotting the Melton index (R) against

the fan gradient (Fg). Based on descriptive statistics, authors found a lower threshold value of 0.75 of the Melton index for debris flows and a threshold value of 7.5° for the fan gradient. Wilford et al. (2004) [26] used a statistical analysis approach on two groups of watersheds, previously discriminated by hydrogeomorphic processes and field recognition of fan deposits. They consider the Melton index (R) and the watershed length (Wl) as the best parameters in discriminating debris flow and debris flood in west-central British Columbia of Canada; the thresholds of these different processes were reported in a scatter plot diagram. In particular, flood watersheds have a Melton index (R) lower than 0.3 and debris flows watersheds a value higher than 0.6. Moreover, according to Wilford et al., 2004 [26], watershed length values lower than 2.7 km are prone to debris flow processes. Santangelo et al. (2006) [27] identified debris flow and fluvial flood processes in the eastern side of the Vallo di Diano through a field survey of the fans and a computation of the morphometric parameters. They plot in a diagram the Melton index against the fan gradient and used the equation of a straight line, as suggested by D'Agostino (1996) [29], to distinguish the boundary of debris flow and fluvial flood processes. In our paper, the morphometric parameters extracted from the selected mountain drainage basins located in the western side of the Vallo di Diano basin, in Southern Italy (Figure 1), are firstly plotted into the three diagrams of De Scally and Owens (2004) [24], Wilford et al. (2004) [26], and Santangelo et al. (2006) [27], are secondly compared with the author's data, and are then discussed, also using the information coming from field investigations. Finally, our investigated area affected by runouts of debris flow processes activated in 1993, 2005, 2007, and 2017 rainfall events have been overlapped on 1:5000 scale contour maps.

### 2.2. The Flow-R Model

The automatic evaluation of debris flows propagation and its susceptibility at a regional scale is mainly based on the use of simplified spatially distributed models [30]. The models use empirical or semi-empirical approaches that have provided to be useful [8,9]. Empirical parameters can be used for the model calibration related to small areas in which one or more past events occurred. However, the improvement of this approach requires the computation of field characteristics on each site [31]. The evaluation of debris flows susceptibility in mountain drainage basins through the identification of potential source areas and the corresponding areas of propagation have been realized through the Flow-R modelling. It is a spatially distributed empirical model developed under Matlab® by the University of Lausanne [11]. According to Horton et al. (2013) [11], the Flow-R application includes two steps, based on the use of a digital elevation model (DEM), as follows: (1) the identification of the potential source area by morphological and user-defined criteria; (2) the assessment of the debris flow propagation based on frictional laws and flow direction algorithms. Note that, the two steps of the application do not evaluate the total amount of debris flows volume and mass in the case of large, investigated areas because excessive mass changes are involved in the erosion and deposition processes [32].

### 2.3. Identification of Potential Source Areas

The identification of the source areas using DEMs is based on a combination of morphological and user-defined datasets. According to Horton et al. (2013) [11], the grid cells of DEMs are classified as (1) favorable, when initiation is possible, (2) excluded, when initiation is unlikely, or (3) ignored, when no decision can be taken on this parameter. According to Rickenmann and Zimmermann (1993) [33] and Takahashi (1981) [34], three morphological parameters can be considered critical for the activation of debris flow, the terrain slope or gradient, the water input, and the sediment availability. The two first parameters are directly extracted using the DEM in the slope and flow accumulation maps, whereas the latter can be manually computed or extracted by DEM using information such as curvature [15]. According to Rickenmann and Zimmermann (1993) [33] and Takahashi (1981) [34], debris flows occur when the slope gradients are higher than 15° and this is the lower initiation threshold used by Horton et al. (2013) [11] in the application of the model.

Furthermore, the plan curvature tool computed using DEMs [35,36] furnishes details about the delineation of the source area affected by a debris flow located in channels. A reference threshold in detecting gullies using the plan curvature by 10-meter-resolution DEM has been found at $-2/100$ m$^{-1}$ [11], although values of $-1.5/100$ m$^{-1}$ and $-0.5/100$ m$^{-1}$ are considered more accurate [17]. It is important to outline that the value of the expected thresholds changes with the location and the typology of the debris flow and with the DEM resolution. The upslope contributing area is calculated by the flow accumulation in GIS software, usually used in hydrological models [37,38], and represents the water input parameter of the Flow-R. Although there is no convergence on the minimum value of the contributing area for the start of the debris flow process, some research analyzing past events in the central Alps has demonstrated an acceptable threshold at 0.01 km$^2$ [11]. The relationship between the upslope contributing area and the slope angle or gradient firstly found by Rickenmann and Zimmermann (1993) [33] and Heinimann (1998) [39] was implemented by Horton et al. (2008) [12] in two curves that allow for the detection of the lower threshold for debris flow source initiation. The curves are referred to as rare and extreme debris flow events and set the slope threshold at 15° for upslope areas larger than 2.5 km$^2$ [34]. The thresholds for rare and extreme events are given in the following Equations (1)–(4), respectively [12]:

$$\tan \beta_{thres} = 0.32 S_{uca}^{-0.2} \; if \; S_{uca} < 2.5 \; \text{km}^2 \tag{1}$$

$$\tan \beta_{thres} = 0.26 \; if \; S_{uca} \geq 2.5 \; \text{km}^2. \tag{2}$$

$$\tan \beta_{thres} = 0.31 S_{uca}^{-0.15} \; if \; S_{uca} < 2.5 \; \text{km}^2 \tag{3}$$

$$\tan \beta_{thres} = 0.26 \; if \; S_{uca} \geq 2.5 \; \text{km}^2. \tag{4}$$

where $\tan\beta_{thres}$ is the terrain slope threshold and $S_{uca}$ is the surface of the upslope contributing area.

### 2.4. Assessment of Debris Flows Propagation

The assessment of the debris flow propagation area, in the Flow-R modelling, is based on two algorithms controlling the path of the debris flow and the runout distance, respectively [12]. In the first case, the path and spreading of the debris flow can be estimated using several flow direction algorithms, which are implemented in the software according to Horton et al. (2013) [11]. The algorithms used in the modelling of flow direction are the D8 [40,41], the D∞ [37], the Rho8 [42], and the multiple flow direction approach [43,44]. The Holmgren's (1994) [45] algorithm, and its modified version by Horton et al. (2008) [12], was mainly used by authors for the susceptibility assessment because it provides most of the other flow direction algorithms and allows a parametrization of the spreading [11], showing those between four and six as optimal values and based on the following Holmgren's equation:

$$p_i^{fd} = \frac{(\tan \beta_i)^x}{\sum_{j=1}^{8} (\tan \beta_j)^x} \tag{5}$$

where $i$ and $j$ are the flow directions, $p_i^{fd}$ is the susceptibility proportion in direction $i$, $\tan \beta_i$ is the slope gradient between the central cell and the cell in direction $i$, and $x$ is the exponent. The modified version of Holmgren's algorithm consists of a change of height in the central cell by a factor named dh, which will change the gradient values. The change provides a smoothing of the DEM roughness and the creation of more consistent spreading values [11,12]. The assessment of the behavior of the inertia, a persistence function, based on Gamma (2000) [46], was implemented by the authors, representing a weighting of the change in the direction concerning the previous direction [12]. In the model, the following three implementations of the persistence have been chosen: the proportional, the cosine, and the Gamma (2000) [46] (see [11,12] for details). Combining the flow direction algorithm

and the weighting of the persistence the following equation was proposed by Horton et al. (2008) [12] to obtain the resulting probability of debris flow spreading as follows:

$$p_i = \frac{p_i^{fd} p_i^p}{\sum_{j=1}^{8} p_j^{fd} p_j^p} p_0 \tag{6}$$

where $i$ and $j$ are the flow directions, $p_i$ is the susceptibility value in direction $i$, $p_i^{fd}$ is the flow proportion based on the flow direction algorithm, $p_i^p$ is the flow proportion based on the persistence, and $p_0$ is the susceptibility value of the central cell previously computed. Using this equation, the spreading, related to a given source, is processed once, computing all the possible paths and attributing probabilities to every cell [11]. The implementation of the runout distance assessment in the model is based on simple frictional laws excluding the source mass and considering unitary the energy balance and is expressed by two algorithms, the two parameters friction model [47] and the simplified friction-limited model (SFLM). The first model, based on a non-linear friction law, is represented by the following equation:

$$V_i = \left( a_i \omega (1 - exp b_i) + V_0^2 exp b_i \right)^{1/2} \tag{7}$$

with $a_i = g(\sin \beta_i - \mu \cos \beta_i)$ and $b_i = \frac{-2L_i}{\omega}$, $V_i$ is the flow velocity at the end of the segment, $\mu$ is the friction, $\omega$ is the mass-to-drag ratio, $\beta$ is the slope angle of the segment, $V_0$ is the velocity at the beginning of the segment, $L_i$ is the length of the segment, and $g$ is the gravity acceleration. However, a correction of the above equation has been made by the authors when the terrain slope decreases [11]. The simplified friction-limited model (SFLM) based on the maximum runout distance reached by the debris flow along its path is expressed by the following equation:

$$V_i = min\left\{ \sqrt{V_0^2 + 2g\Delta h - 2g\Delta x \tan \varphi}, \ V_{max} \right\} \tag{8}$$

where $V_i$ is the flow velocity at the end of the segment, $\Delta h$ is the difference in elevation between the central cell and the cell in direction $i$, $V_0$ is the velocity at the beginning of the segment, $g$ is the gravity acceleration, $\Delta x$ is the increment of horizontal displacement, $\tan\varphi$ is the gradient of the energy line, and $V_{max}$ is the velocity limit. Based on the maximum velocity of debris flow observed in Switzerland by Rickenmann and Zimmerman (1993) [33], a velocity value of 15 ms$^{-1}$ was adopted in the model [11,12].

The processing of a susceptibility assessment requires four levels of matrices corresponding to in-sequence levels of propagation and named as (i) processing level, (ii) active cell list, (iii) current propagation, and (iv) result layer [11]. Finally, as result, the model proposes three different types of implemented runs that require a very different time of processing. They are the overview type that triggers only the sources placed at the top of catchments, the quick type that triggers all the source cells and propagate the remaining sources in a subsequent stage, and the complete type that triggers all the source areas and does not realize a check on previously processed propagations [11]. In addition, any other dataset can also be added in the input data of the Flow-R software, such as geological maps to account for sediment availability, or land use maps helping to remove inaccurate source areas. In the flow chart of Figure 2, the methodological steps of the process within the Flow-R software are schematically synthetized as proposed by Horton et al. (2013) [11].

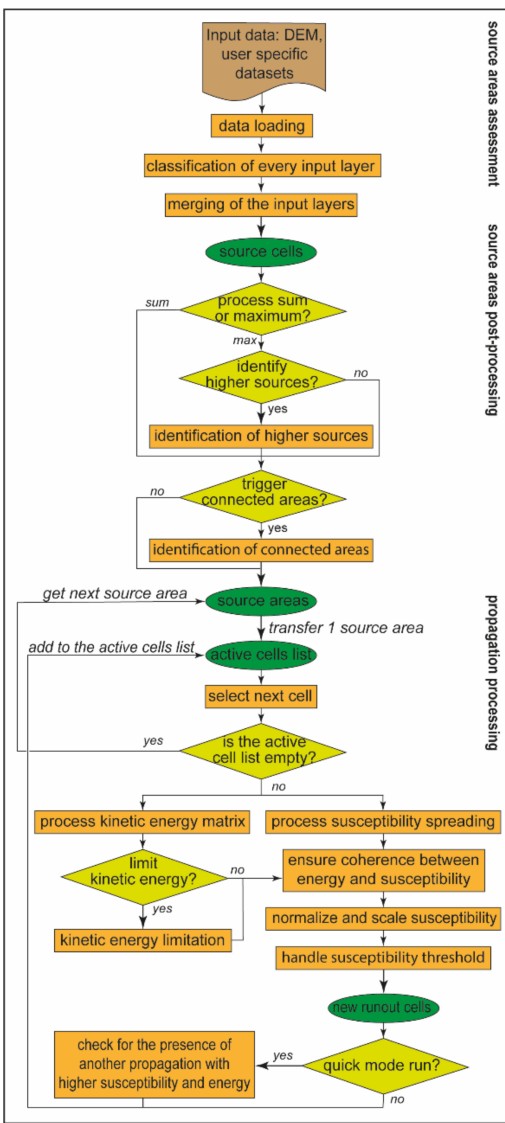

**Figure 2.** Flow diagram of the processes in the Flow-R software as proposed by Horton et al. (2013) [11] (modified by [11]).

The mapped areas of the runout events affecting some selected watershed basins have been used in comparison for the modelling areas, thus providing the best fit parameters applied in the Flow-R modelling. A qualitative, visual inspection was produced by a superposition of the real rainfall events recorded in 1993, 2005, 2010, and 2017 and the runout maps resulting from the modelling process. The comparison has been conducted following an iterative process of the superposition of different modelling maps. The choice of the best parameters, used in the Flow-R modeling, follows the iterative procedure in which they are modified until the output map fits well with the real extension of the observed debris flows. Then, the best-fit map procedure was extended to the whole western side of the Vallo di Diano basin. The output maps were extracted from a 5-m-resolution DEM using the free available QGis software and were implemented in the Flow-R software. The slope angle threshold of the sub-basin's upstream area was selected following the value explained in the model description, which considers the best fit scenario for debris flow activation.

## 3. Geological and Geomorphological Settings

### 3.1. Regional Framework

The Southern Apennines are a north-east verging fold-and-thrust belt generated by a Neogene deformation of the African palaeomargin ([48], and references therein), strongly dismembered by Quaternary neotectonics (Figure 1). Starting from Late Miocene, the orogen underwent a low-angle extension that led to the exhumation of its non-metamorphic core complex constituted of Mesozoic Lagonegro-type pelagic units [49]. Transpressional to transtensional tectonics was responsible for the Pliocene to Early Pleistocene evolution of the southern-Apennine chain, whereas a high-angle extensional faulting took place in mid-Pleistocene times in the axial zone of the chain [50–52]. The southern Apennines chain can be roughly divided into three sub-parallel main zones, inner, axial, and outer, respectively (a, b, and c in Figure 1). The inner belt corresponds to the Tyrrhenian side of southern Apennines, and includes Cretaceous-to-Miocene, deep-water, pelagic successions (Liguride and Sicilide units [53]) over-thrusted on Mesozoic, shallow-water carbonates units [54] forming the Cilento Promontory, with a maximum elevation reached at Cervati Mt. (1900 m a.s.l.). Transverse valleys are cut by rivers flowing toward the Tyrrhenian Sea, locally producing narrow gorges as deep as ca. 100 m. Moreover, some tectonically controlled intermontane basins are placed in the more eastern zone [55,56].

The axial belt is made up of Mesozoic-to-Cenozoic, shallow-water carbonate units (Apennine Platform) over-thrusted on top of coeval deep-water, pelagic carbonates and siliciclastics units (Lagonegro Units [48,57]). The flysch sequences originally deposited in the Miocene foredeep basin and clastic deposits of Pliocene thrust-top basins were both involved in contractional deformation, and rest unconformably on top of the tectonic units. In the axial zone, the highest mountains peaks are up to 2000 m a.s.l. and correspond to remnants of the Pliocene to Pleistocene erosion land surfaces uplifted and dismembered by Quaternary faults [58,59]. The landscape includes block-faulted mountains bounded by fault scarps retreated by slope replacement processes [60,61], and fault-bounded intermontane basins filled with fluvial and lacustrine deposits [55,56].

The outer zone is formed by Cenozoic sandstone-to-clayey units that are buried eastward underneath the Pliocene-to-Pleistocene clastic deposits of the Bradano Trough [48] and Quaternary volcanics [62]. This zone is characterized by a general, north-eastward tilting of the units due to the uplift of the axial zone of the chain, and displays a landscape with an elevation up to ca. 1100 m a.s.l. There, the fluvial valleys form a wide apron trending NE–SW, in the northern sector, to E–W, in the southern one [63].

In conclusion, the regional setting allowed then to know what the main lithologies are and what the most relevant areas that can be potentially involved in debris flow processes in a frame of mountain chain such as the southern Apennines are.

### 3.2. Geology and Geomorphology of the Vallo di Diano Basin

The Vallo di Diano intermontane basin is located in the southern sector of the Campania Region of Italy and represents the biggest Pleistocene morphotectonic basin of the southern Apennines' chain [56]. It has an area of about 720 km$^2$, a length of about 35 km, and is developed from NW to SE according to the regional elongation of the southern Apennine chain. The basin is bounded by the Maddalena ridge in the northeast flank and by the Alburno-Cervati ridge in the northwest side. A N140-150°-striking normal master fault bounds the eastern side of the basin (Figure 3), whereas the western flank is affected by N120°-trending left-lateral strike-slip faults and by NW–SE-oriented Apennines faults [64]. The eastern sector of the basin is bounded by the Maddalena Mts, a NW–SE-elongated ridge, mainly composed of Mesozoic to Cenozoic platform carbonates of the Mt. Marzano-Mt. della Maddalena tectonic Unit (Figure 3), over-thrusted on the coeval deep-sea water carbonate and siliciclastic successions of the Lagonegro tectonic Unit [48,57]. The western sector of the Vallo di Diano basin is bounded by the La Marta, Cocuzzo delle Puglie, Motola, Cervati Mts., mainly composed of Mesozoic to Cenozoic platform carbonates pertaining to a different tectonic unit named as Alburno-Cervati-Pollino

tectonic units [48,57] (Figure 3). Miocene terrigenous units are also scattered in the whole two sectors and were involved in the contractional deformation. The sedimentary infill of the basin is formed of fluvio-lacustrine deposits and coeval slope-to-alluvial fan deposits placed along the flanks of the basin (Figure 3). It is composed of two synthem units, Early Pleistocene to Holocene in age, separated by a first order unconformity bound [65].

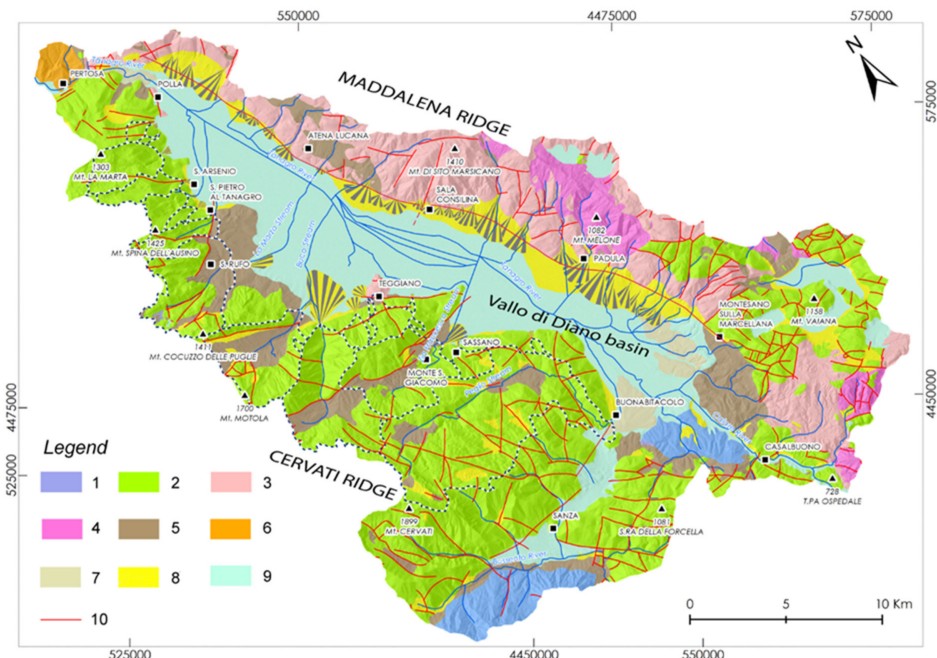

**Figure 3.** Geological sketch map of the Vallo di Diano intermontane basin. Legend: (1) Cretaceous to Oligocene internal units; (2) Mesozoic-Cenozoic carbonates of the Alburno-Cervati-Pollino unit; (3) Mesozoic-Cenozoic carbonates of the Marzano-Maddalena unit; (4) Lower–Middle Triassic to Oligocene siliceous and carbonate successions of the Lagonegro units; (5) Miocene siliciclastic units; (6) Pliocene clastic unit; (7) Pleistocene lacustrine deposits; (8) Pleistocene coarse-grained deposits; (9) Holocene fluvio-lacustrine deposits; (10) faults. Dotted lines are divides of mountain drainage basins of the western side valley.

The study area is characterized by a NE exposure and the climate is Mediterranean, humid, temperate with a mean annual rainfall variability that ranges from 1000 to 2000 mm, and maximum monthly values of 160–190 mm [66]. The basin flanks are featured by several orders of land surfaces, ranging in age from late Pliocene to middle Pleistocene [67] and by many fault scarp features responsible for the formation of the trough. The latter is arranged in three different depositional depocenters filled by fluvio-lacustrine successions, as revealed by seismic data [68]. Karst landforms are widely distributed in the whole carbonate massifs bounding the floodplain floor as a consequence of the planation activity mainly due to karst contact processes. In the western side valley, in the Cervati Mt. area, large relicts of palaeosurfaces are found and sculptured on both carbonate rocks and terrigenous units [55,64]. The high present day relief is the result of both the tectonic uplift of the area and the related vertical fluvial incision. The latter sometimes was responsible for the entrenchment of streams by a geomorphological superposition mechanism. This is the case of the Sammaro Stream [69] and the Buccana Stream [67]—both pertaining to the Motola ridge—that operated a fluvial vertical incision at a rate of 0.2–0.3 mm/y during middle Pleistocene times. All the two catchment stream basins were developed in the Mt. Motola Ridge, a carbonate-dominated morphostructure that is 18 km long and about 2 km large (Figure 3).

The Mt. Motola Ridge is a narrow N110-120°-trending fold displaced by high-angle faulting in the middle to late Pleistocene during the formation of the Vallo di Diano trough [56,67]. It was interpreted by Putignano and Schiattarella (2008) [67] as a strike-slip duplex generated within a left-shear zone. In particular, the Buccana Stream transversally cuts the southern sector of the Mt. Motola Ridge, thus producing a NE–SW-oriented deep gorge vertically incised in the Mesozoic to Cenozoic platform carbonate of the Alburno-Cervati-Pollino Unit (Figure 3). It represents an antecedent stream produced by the recent uplift of the western side of the Vallo di Diano basin [65,67,69]. In particular, the Buccana fluvial catchment is formed by the following three distinct geomorphological units: the first is a small sub-basin located in the upper reach that was captured by the headwater erosion of the Buccana Stream, the second is a deep antecedent gorge valley incised by the Buccana Stream in carbonate bedrock, and the third is the depositional fan area produced at the toe of the stream valley. In conclusion, the local geological arrangement of the western side of the Vallo di Diano basin has revealed a similar lithological composition of the bedrock mainly composed of carbonate rocks. Considering the low susceptibility of these rocks to debris flow, the main geological factor in triggering the process is the thickness of the recent slope deposits. Furthermore, the morphology of the study area has shown the existence of deep incised valleys that are the preferential ways of flow.

## 4. Results

### 4.1. Morphometric Parameters

Along the western side of the Vallo di Diano intermontane basin, 19 mountain drainage basins were selected and numbered from B1 to B19, starting from the northwest to southeast (Figure 4 and Table 2). A first step conducted by means of aerial photos interpretation at a 1:33.000 scale has revealed that not all the watershed basins have generated the related fan landform at the piedmont valley. It is the case of the Peglio drainage basin (B19 in Figure 4) that does not show a fan landform, nevertheless it represents the largest watershed basin of the western side of the Vallo di Diano. The lacking of a fan in this basin can be attributed to the following two factors: (1) the absence of sediment yield due to the lower water discharge because of the water flow is, quite exclusively, on carbonate rocks that have high infiltration rates; (2) the existence of a recent fluvial capture of the Peglio Stream (Figure 4) that transports the sediment yield in an entrenched valley that, at Buonabitacolo, runout all the sediments in the Tanagro River. Consequently, the morphological analysis of the aerial photos allowed us to detect 18 fan landforms that are investigated in the field and then are numbered from F1 to F18 (Figure 4 and Table 3).

The Peglio Stream reaching about 50 km$^2$ of width has also both the maximum elevation among all basins at 1896 m a.s.l. and the most relief of about 1396 m. Conversely, the smallest one is the Vallone del Duca basin (B14 in Figure 4) reaching about 0.18 km$^2$ of width, a max elevation of 913 m a.s.l. and a relief of 430 m (Table 1). The average dimension of the 19 watersheds is about 7.97 km$^2$ and, excluding the widest Peglio basin, it decreases to about 5.63 km$^2$, which seems to be more correspondent to the real width of the western side watersheds. According to the literature, the lower threshold of the contributing area in the starting of debris flow process is about 10 km$^2$ in the Central Alps [33,70], reaching lower values in Norway [17].

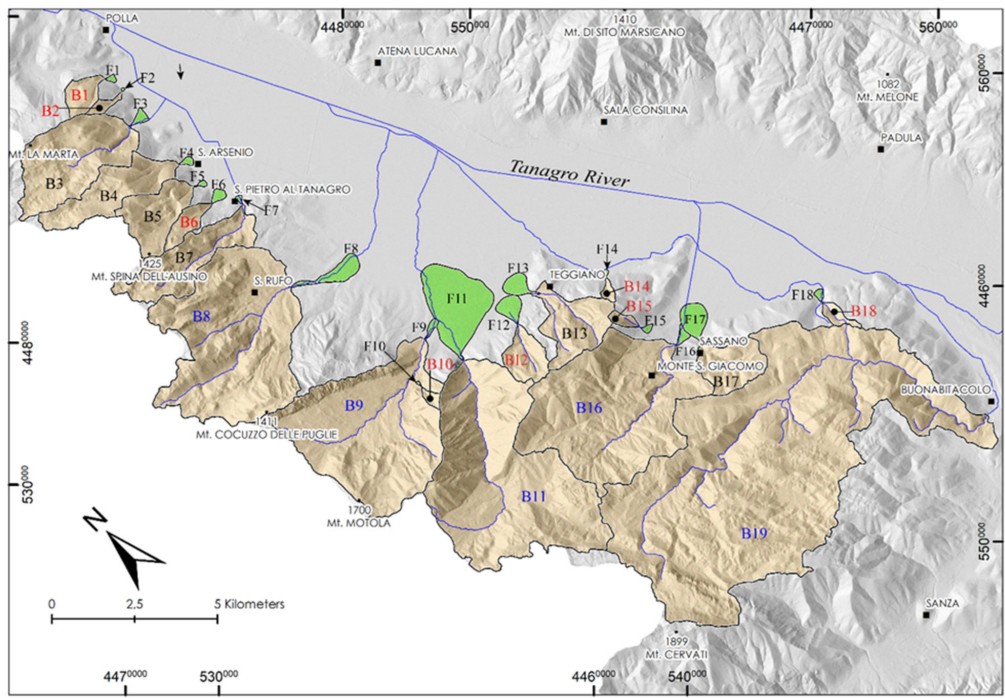

**Figure 4.** Mountain drainage basins (brown) and related fans (green) in the western side of the Vallo di Diano intermontane basin. The B acronym indicate drainage basins width: reds are smaller than 2.5 km², blacks are included between 2.5 and 10 km², and blues are larger than 10 km². The F acronym indicates the related fan landforms.

**Table 2.** Morphometric parameters of the mountain drainage basins of the western side of Vallo di Diano intermontane basin.

| Basin N. | Name | Hmax (m) | Hmin (m) | Hmean (m) | Br (m) | Ba (km²) | Wl (m) | R |
|---|---|---|---|---|---|---|---|---|
| B1 | V. Pastena | 1127.7 | 484.7 | 769.9 | 643 | 0.97 | 1611 | 0.65 |
| B2 | V. Petrosa | 992.9 | 479.8 | 785.6 | 513.1 | 0.27 | 1263 | 0.98 |
| B3 | V. Tuorchi | 1301.6 | 488 | 994.8 | 813.6 | 6.97 | 4416 | 0.31 |
| B4 | V. Tornaturi | 1332.4 | 489.7 | 1033.2 | 842.7 | 3.97 | 4569 | 0.42 |
| B5 | T. Futorella | 1426.9 | 951.6 | 1003.8 | 475.3 | 3.0 | 2664 | 0.27 |
| B6 | F.so della Torre | 1312.6 | 481.9 | 893.9 | 830.7 | 0.85 | 2051 | 0.9 |
| B7 | V. Setone | 1468.6 | 461 | 1042 | 1007.6 | 4.65 | 4725 | 0.47 |
| B8 | T. Marza | 1471.3 | 483.8 | 915.1 | 987.5 | 17.5 | 5491 | 0.24 |
| B9 | T. Buco | 1735.6 | 505.7 | 1053.6 | 1229.9 | 16.2 | 5660 | 0.3 |
| B10 | Valle Torto | 1273.2 | 681.5 | 1035.7 | 591.7 | 0.2 | 932 | 1.3 |
| B11 | T. Buccana | 1738.1 | 544.2 | 1101.7 | 1193.9 | 21.0 | 7203 | 0.26 |
| B12 | Valle Cupa | 1410.8 | 572.3 | 1068.1 | 838.5 | 1.58 | 1808 | 0.67 |
| B13 | V. Sinagoga | 1390.6 | 483.5 | 959.7 | 907.1 | 3.39 | 2555 | 0.5 |
| B14 | V. del Duca | 913.3 | 711 | 756.7 | 430.3 | 0.18 | 798 | 0.99 |
| B15 | V. Secco | 1114.1 | 404.3 | 959 | 403.1 | 0.27 | 999 | 0.76 |
| B16 | T. Zia Francesca | 1445.3 | 497.6 | 966.1 | 947.7 | 17.1 | 5773 | 0.23 |
| B17 | V. San Nicola | 924.3 | 506.9 | 741.5 | 417.4 | 2.52 | 2086 | 0.26 |
| B18 | T. Valla | 828 | 475.1 | 663.7 | 352.9 | 0.42 | 1429 | 0.54 |
| B19 | T. Peglio | 1896.6 | 500.3 | 1008 | 1396.3 | 50.30 | 12,134 | 0.19 |

**Table 3.** Morphometric parameters of the fans in the western side of Vallo di Diano intermontane basin.

| Fan N. | Fan Area (km$^2$) | Fan Apex Elevation (m) | Fan Toe (m) | Fan Gradient (Degree) |
|---|---|---|---|---|
| F1 | 0.048220 | 484 | 448 | 5.6 |
| F2 | 0.009039 | 475 | 448 | 10.8 |
| F3 | 0.134259 | 487 | 450 | 3.8 |
| F4 | 0.074192 | 490 | 448 | 5.2 |
| F5 | 0.040244 | 476 | 448 | 3.9 |
| F6 | 0.128934 | 480 | 446 | 3.6 |
| F7 | 0.032508 | 465 | 457 | 1.5 |
| F8 | 0.655337 | 486 | 455 | 0.8 |
| F9 | 0.096612 | 508 | 487 | 1.9 |
| F10 | 0.010876 | 682 | 640 | 12.9 |
| F11 | 3.686782 | 560 | 465 | 1.8 |
| F12 | 0.432525 | 572 | 464 | 6.8 |
| F13 | 0.423579 | 505 | 455 | 2.7 |
| F14 | 0.026371 | 483 | 449 | 9.0 |
| F15 | 0.057295 | 707 | 593 | 17.5 |
| F16 | 0.124157 | 501 | 480 | 1.9 |
| F17 | 0.660485 | 508 | 465 | 1.9 |
| F18 | 0.071381 | 475 | 462 | 1.9 |

In the Flow-R modelling, Horton et al. (2008) [12] proposed lower values for debris flow source activation and related a value of watersheds smaller than 2.5 km$^2$ to both rare and extreme rainfall events. They follow the morphometric parameters previously suggested by Rickenmann and Zimmermann (1993) [33] and by Heinimann (1998) [39].

The first category includes five watersheds larger than 10 km$^2$; the second, six watersheds ranging from 2.5 to 10 km$^2$; and the third one, eight watersheds smaller than 2.5 km$^2$ (Figure 4 and Table 2 for details).

Considering that the contributing area of watersheds is almost formed of Mesozoic to Cenozoic carbonate bedrock and few areas accommodate terrigenous units, it is possible to consider all the basins substantially homogeneous from a lithological point of view. In this regard, if considering the relief value of basins—that represent, among others, a parameter suggesting the potential capability to erode a basin—then it can be plotted versus the basin areas, thus furnishing information about the potential erodibility in the watersheds. A log–log plot of the basin relief and basin area shows a direct straight line behavior, meaning that a growth of the relief produces a correspondent growth of the basin areas (Figure 5), as expressed by the power law in Figure 4, which could be implemented for the study area. As a result, watersheds larger than 10 km$^2$ can be affected by higher potential erodibility than the smaller ones (<2.5 km$^2$) and the difference in erodibility could be due to debris flows or fluvial/debris floods.

Basin area parameter was also used in the Canadian Cordillera to identify debris flow torrent basins that are characterized by an upper threshold of about 10 km$^2$ [25] and in the southern Alps of New Zealand identifying debris flow watersheds with an area lower than 4–5 km$^2$ [24]. According to this classification, we could preliminarily classify as watershed prone to a debris flow process those pertained to the second (ranging from 2.5 to 10 km$^2$) and third (<2.5 km$^2$) categories by assigning the fluvial flood process to the first category of watersheds larger than 10 km$^2$ (Table 2).

The scientific literature on the susceptibility assessment of alluvial fans in the eastern side of the Vallo di Diano basin is based on the historical damages of urbanized fans that produced local hazard maps distinguishing four different classes, from low to high degree [22,27,71]. The authors extract several morphometric parameters from the watersheds and related fans formed at Maddalena Ridge's foothill, Vallo di Diano floor basin's eastern side, including the Melton index and the fan gradient. The values of these two indices were plotted in a diagram, allowing the authors to identify the two main hydrogeomorphic processes classified as debris flow/flood and fluvial [27] (Figure 6a).

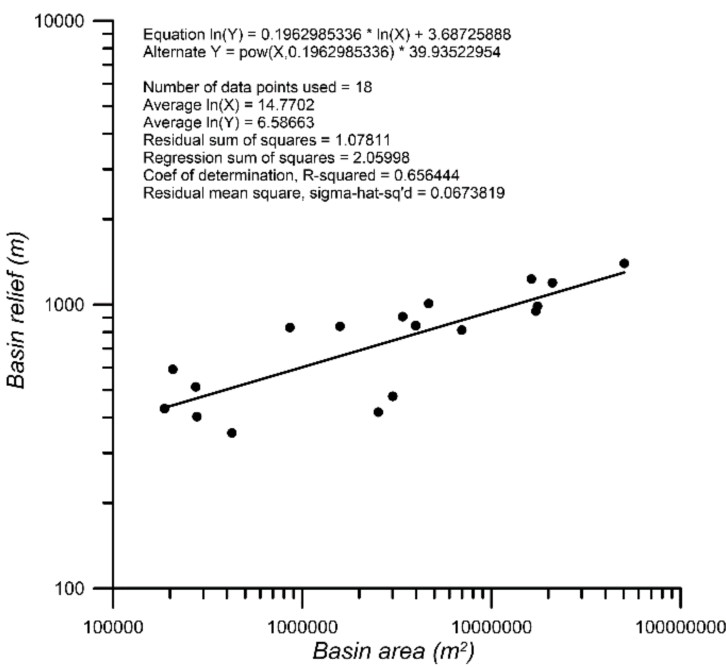

**Figure 5.** Plot of the basin area versus basin relief. See text for details.

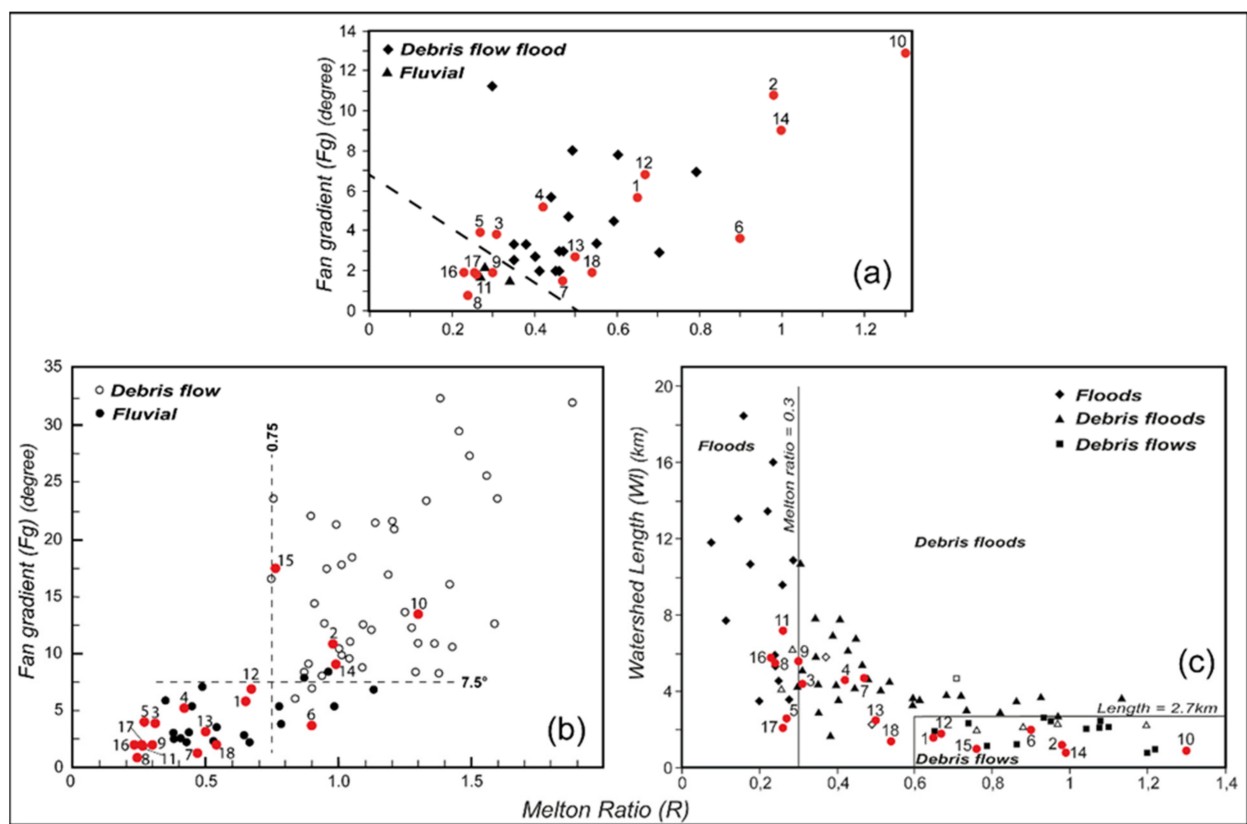

**Figure 6.** Diagrams plotting the Melton ratio (R) versus the fan gradient (Fg) or the watershed length (Wl). (**a**) Redrawn from Santangelo et al., 2006 [27]; (**b**) redrawn from De Scally and Owens (2004) [24]; (**c**) redrawn from Wilford et al., 2004 [26]. Red circles are plots of the 19 mountain drainage basins of the study area.

We point out that both the meaning of these processes in physical terms and the Melton ratio was explained in the previous Section 2.1 "Morphometric parameters". With the aim to verify a possible correspondence, we plotted our data of the Vallo di Diano western flank into the diagram elaborated by Santangelo et al. (2006) [27]. As a result, 14 watershed basins of the western side of the Vallo di Diano are in the debris flow/flood field and five in the fluvial one (Figure 6a). In the southern Alps of New Zealand, the morphometric parameters of the Melton index (R) and the fan gradient (Fg), together with descriptive statistics were used by De Scally and Owens (2004) [24] in order to discriminate the threshold value separating debris flow and fluvial flood processes (Figure 6b). Adding the Vallo di Diano data in the author's plot diagram, we observe that only four watersheds are into the debris flows field, whereas 15 basins are in the fluvial flood area (Figure 6b). In the Canadian Rocky Mountain, Wilford et al. (2004) [26] plotted the Melton index in combination with the watershed length and showed a discrimination area between the watersheds prone to debris flows and debris floods (Figure 6c). Additionally, in this case we add the Melton index and the watershed length related to the 19 watersheds of the study area into the scatterplot of Figure 5c. We found that seven basins are in the debris flow area, characterized by a Melton index >0.6 and a length lower than 2.7 km. Furthermore, five basins are included in the debris floods area and five basins in the fluvial flood area (Figure 6c). The basin nine cannot be attributed because it is on the threshold of the two latter processes indicated by the vertical straight line with a Melton index of 0.3.

A merge of information coming from the three different diagrams above presented, provides, in a preliminary and very fast way, a distinction between the watersheds falling within the area prone to the debris flow process and those falling within the area related to fluvial processes. As a result, we observe that only four basins prone to debris flows process are common into the three diagrams of Figure 6 and precisely the B2, B10, B14, and B15 watersheds. The three basins B1, B6, and B12 are only included in the distinctive debris flow area of the two diagrams of [24,26]. Finally, seven watersheds are in the diagram of [27]. The high number of basins in the latter diagram (Figure 6a) is readable considering that the data come from the opposite flank of the same study area (the eastern side of the Vallo di Diano) that features a similar landscape and then comparable morphological parameters.

### 4.2. Calibration of the Flow-R Methodology

The calibration of Flow-R parameters that might be used in the modelling process has requested a back-analysis of the propagation runout area at a local scale. Within the analyzed basins, the following fluvial catchments affected by real debris flow/flood events were selected as case studies: (1) the Valle Torto (B10), (2) the Buccana (B11), (3) the Valle Cupa (B12), (4) the Sinagoga (B13) watershed basins, and further, (5) small watershed included within the Marza basin (B8) (Figure 4). The strong rainfalls that occurred in the years 1993, 2005, 2010, and 2017 in the Vallo di Diano area produced runout events drawn in four thematic maps at different scales related to the Buccana (Figure 7a), Valle Cupa (Figure 7b), Sinagoga (Figure 7c), and Marza (Figure 7d) watershed basins. The boundaries and propagation areas represented are based on both post-event field observations and an interpretation of aerial photograms (Figure 7).

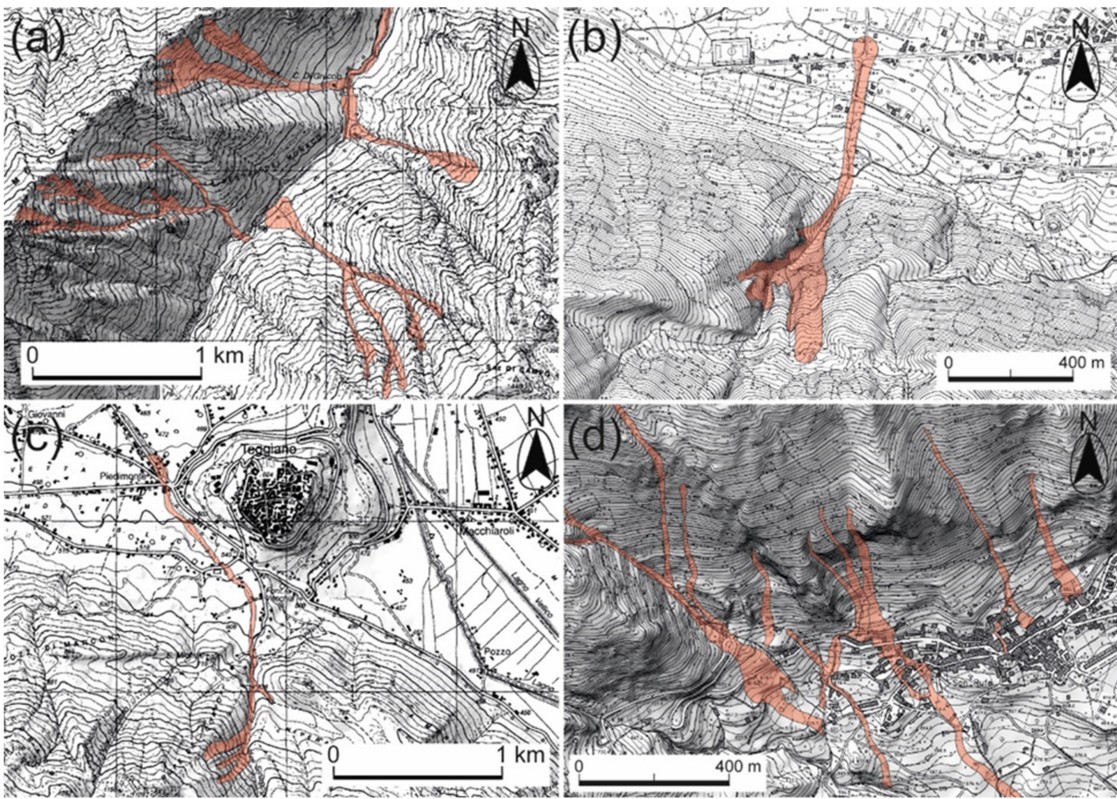

**Figure 7.** Area of runout events occurred in the Buccana (**a**), Valle Cupa (**b**), Sinagoga (**c**), and Marza (**d**) watershed basins during the strong rainfall events of 1993 (**a**,**b**), 2005 (**c**), and 2010 and 2017 (**d**).

Furthermore, field investigations based on the stratigraphical and sedimentological analyses of the deposits furnished further information on the discrimination of debris flows and fluvial flood activities, as shown in Figures 8 and 9. The Buccana watershed basin (B11 in Figure 4) is extended from the mountain top divide, located at Cerasulo and Serra la Corva Mts reaching about 1738 m of elevation a.s.l., to the down valley at the fan apex of San Marco di Teggiano, at about 544 m a.s.l. (Figure 4). The basin is about 21 km² in width and is included in the first (>10 km²) category of watersheds (Figure 6 and Table 2). The Buccana Stream valley is an antecedent fluvial V-shaped valley and, in the lower reach, cuts carbonate rocks [65,67,69]. A large incised alluvial fan is formed at the apex of the valley then favoring the entrenchment of deposits in the channel. At the outlet of the Buccana Stream channel, a debris flood process was observed after the rainfall event of October 1993 and debris flows deposits were observed only into the lateral tributary sub-channels (Figure 7a). Due to the many transversal weirs in the Buccana channel made in 1960, the transport of coarse-grained clastic deposits in the main channels was inhibited. This means that the main debris flow processes were activated along the flank valleys, in the tributary sub-basins, and the deposition was converged into the Buccana channel and stopped out (Figures 7a and 8c). The Valle Cupa watershed basin (B12 in Figure 4) is about 1.58 km² in width and is sourced at about 1410 m of altitude a.s.l. at Serre di Campo Soprano Mt. The outlet of the basin is located at about 572 m a.s.l. and the stream after 330 m of length across the road S.P. 263. The drainage pattern is almost dendritic, and the length of the streams is about 7500 m, with the main channel being 1950 m in length. The Valle Torto watershed basin (B10 in Figure 4) is smaller than the previous one, reaching about 0.20 km² of width, and is sourced from the Motola Mt. top-mountain ridge at 1273 m of elevation a.s.l. Its outlet is located at the confluence of the Rocca Longa Stream at about 681 m a.s.l., the drainage network shows a parallel pattern, and the main channel is about 1400 m in length. Both these two watersheds have an area smaller than 1 km² and pertain

to the third category of basins as interpreted by morphometric parameters (Figure 6 and Tables 2 and 3).

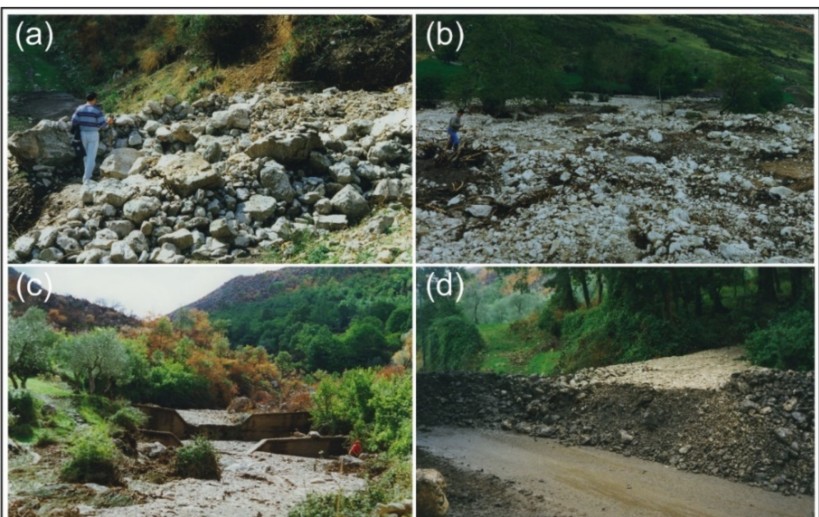

**Figure 8.** Debris flow deposits produced after the rainfall events of 1993 and 2005 in the Buccana (**a**,**c**), Valle Torto (**b**), and Valle Cupa (**d**) sub-basins. In photo (**d**) the debris flow deposit about 2 m thick that was dragged out by the road is shown.

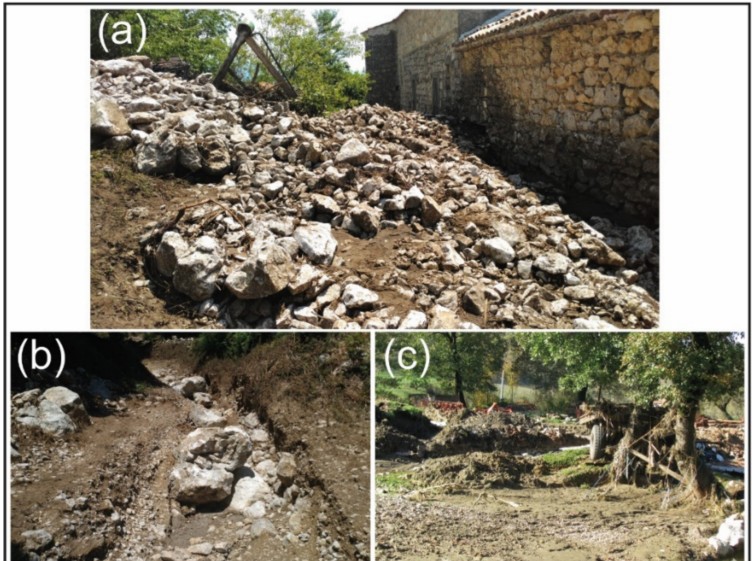

**Figure 9.** Depositional lobes generated after the rainfall events in the Marza sub-basin catchments. (**a**,**b**) are related to the 2017 event and (**c**) is related to the 2010 event. This is a figure. Schemes follow the same formatting.

Both the Valle Cupa and Valle Torto watersheds were affected by debris flows in October 1993 triggered by a rainfall event that produced more than 50-mm high water in a time of two/three hours and then causing an inundation of roads and buildings with detritus (Figure 8). The Sinagoga watershed basin (B13 in Figure 4) is located close to the right-side Valle Cupa basin and, covering an area of about 3.4 km$^2$, belongs to the second category of basins (Figure 6 and Table 2). The source area is located at 1390 m of elevation a.s.l. and the outlet point is at about 483 m a.s.l. (Table 2). The watershed was affected by a debris flow in 2005 that runs mainly small clastic deposits over the outlet point. Four sub-basins smaller than 1 km$^2$ are present in the Marza watershed and flow mainly from north-west to south-east (Figure 9). It is important to outline that the Marza watershed

pertains to the first category of basins and, therefore, to basins affected by fluvial floods because its area is larger than 10 km$^2$ (17.5 km$^2$). Moreover, the source area is at 1470 m of elevation a.s.l. and the outlet is located at about 483 m of elevation a.s.l., reaching a relief of 9875 m (Table 2). Conversely, the four sub-basins that are left tributaries of the Marza basin, reaching about 1238 m of elevation a.s.l. and outlet points distributed around 650 m a.s.l., thus reaching a relief of 588 m (Figure 7d). The drainage pattern in these sub-basins is rectilinear, showing a direct influence of fault segments on their development. Their relevance is related to the fact that the water runouts are directed towards the San Rufo village. The whole area was affected by intense rainfall during 2010 and July 2017, reaching about 180 mm high in 24 h that triggers mass movement processes. Field investigations have revealed that both the debris flow and the debris flood events were the main type of processes. The events involved the northern portion of the village, causing damages to buildings and roads (Figure 9a–c).

### 4.3. Flow-R Parameters

The runout events affecting the Buccana (B11), the Valle Cupa (B12), 547 the Valle Torto (B10), the Sinagoga (B13), and the Marza (B8) watershed basins (Figure 4) have been mapped and used as reference maps in the choice of the best fit parameters (Figure 10). The first step in the modelling process consists in the source area identification, and for this purpose, the thematic maps of elevation, slope, plan curvature, and flow accumulation were produced (Figure 11). Take into account that the latter maps of flow accumulation are not reproduced in the next figures. The abovementioned maps related to the Buccana, the Valle Torto, the Valle Cupa, the Sinagoga, and the Marza watershed basins (Figure 11) were extracted from a 5-meter-resolution DEM using the QGIS software and then were implemented in the Flow-R software. The slope angle threshold was selected according to the best fit scenario for debris flow activation. Many examples are found in different parts of the world, such as the Alps (both Italian and Swiss mountain flanks), Canada, and Japan, identifying a value ranging from a minimum of 7° to a maximum of 20° ([11] and references therein).

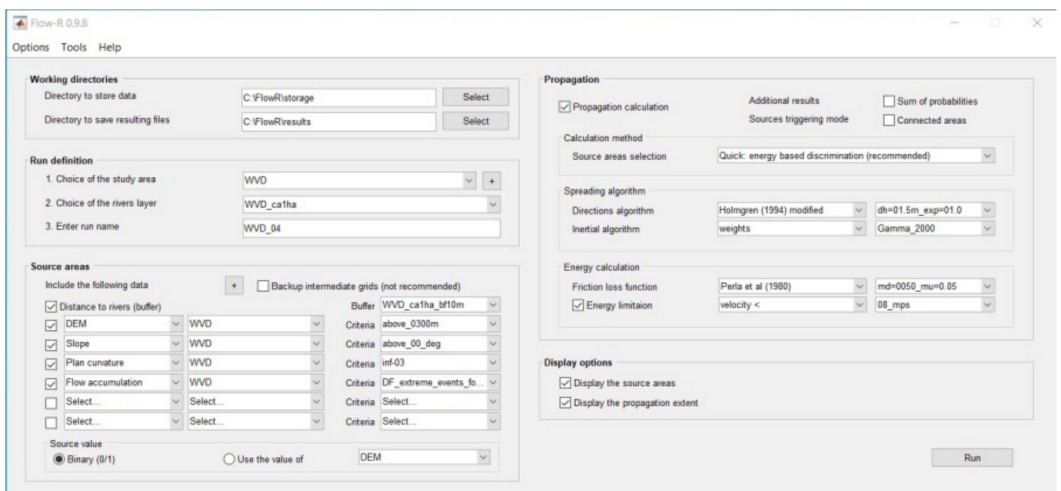

**Figure 10.** Main frame of the Flow-R software used in the implementation of parameters.

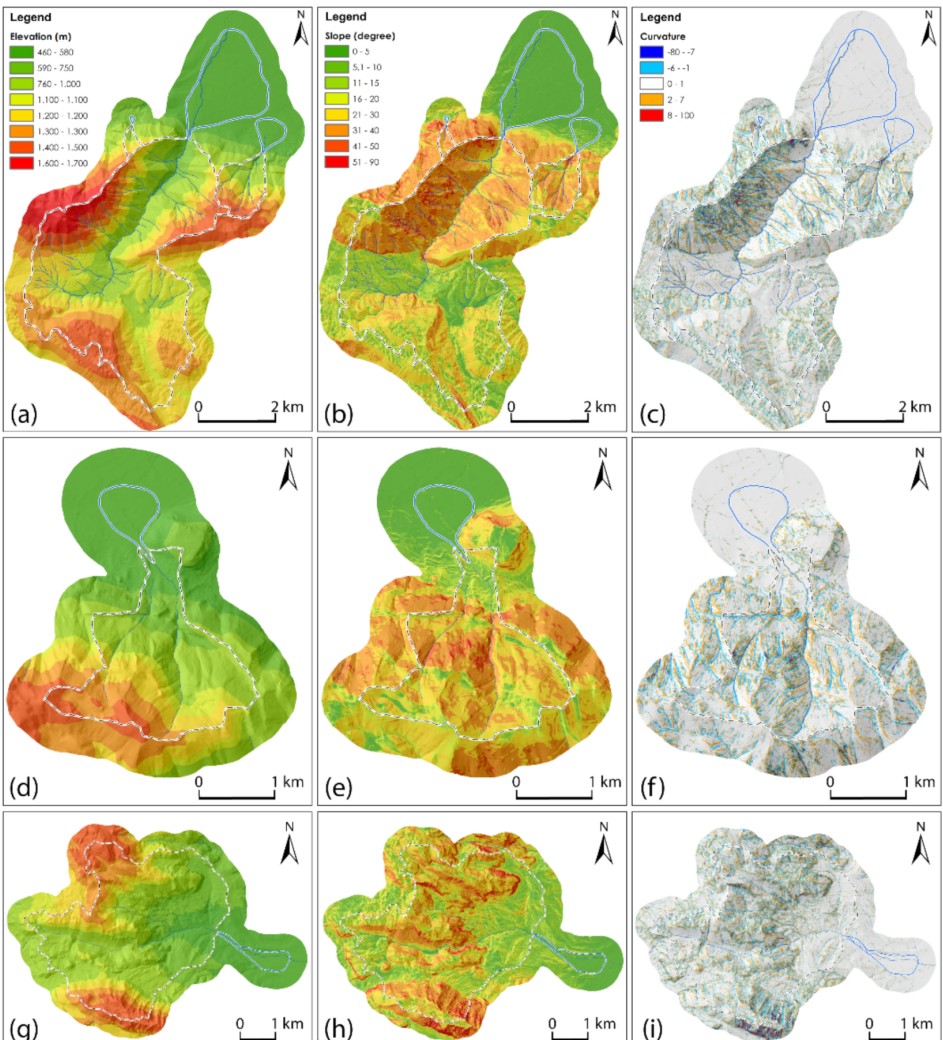

**Figure 11.** Maps of elevation (**a,d,g**), slope (**b,e,h**), and curvature (**c,f,i**) related to the five investigated watersheds. (**a–c**) are referred to Buccana (B11), Valle Torto (B10), and Valle Cupa (B12) basins; (**d–f**) are referred to the Sinagoga (B13) basin; (**g–i**) are referred to the Marza (B8) basin.

According to Horton et al. (2008) [12], the threshold for the slope angle is related to the size of the upslope basins. Upslope areas greater than 2.5 km$^2$ consider a slope threshold of 15° for triggers debris flows, whereas smaller upstream basins can be possible sources of activation if the slope angle is higher than 15°. Following a conservative criterion, an iterative procedure varying the slope threshold from 15 to 20° has been implemented in the modelling. It is known that planar curvature with a negative value allows us to recognize concave slope morphologies such as hollows, gullies, and channels [11]; on the basis of this assumption, we set the threshold at $-2/100 \text{ m}^{-1}$. The geological map of the area has been excluded from the automatic elaboration, considering that the area is quite homogenous from a lithological point of view. In fact, the main rocks cropping out are carbonate rocks, whereas terrigenous rocks are less present (Figure 3). The iterative process of modelling allowed us to choose the best value of 15° as the threshold value for the activation of debris flows processes on the western side of the Vallo di Diano basin. This value fits well with the standard value suggested by many authors [15,33,34].

In the second calibration step, the assessment of the debris flow spreading area has been computed using the modified version of the Holmgren's spreading algorithm [12]. In this step, a 3 × 3 grid cell elevation has been also considered, thus changing the gradient values. The exponent x of the algorithm, which controls the divergence, can increase starting from x = 1, with a spreading similar to the multiple flow direction, to x = ∞, where

a reduction in the divergence leads to a single flow direction. A value of x = 4, allowing the best and wide range of flow accumulation to be reproduced, has been used in the modelling by Horton et al. (2013) [11], following the setting previously proposed by Claessens et al. (2005) [72]. With the aim to reproduce not a single event of propagation but a spreading covering all the possible events in the area, a value of x = 1 and a dh = 1.5 have been considered. Furthermore, the Gamma (2000) [46] persistence function has been used as the inertial parameter in the assessment of the spreading [11]. The Perla et al. (1980) [47] model has been used in the evaluation for the two friction parameters and for the assessment of the runout distance. Horton et al. (2011) [15] suggest μ = 0.09 and M/D = 30 as optimal value parameters both for the Switzerland and Pakistan areas. Conversely, the best fit parameters values we adopted in the western Vallo di Diano are μ = 0.05 and M/D = 50, adding a limitation of velocity at 8 m/s. As a result, the exposed areas prone to debris flow spreading were evaluated in association with a qualitative potential susceptibility of the selected watershed basins (Figure 12A,B). In the map of Figure 12A,B, five categories of potential debris flow activation were discriminated. In particular, the red and orange colors correspond to a high probability of runout spreading, the yellow color to middle probability, and the dark/light blue color are the lower ones. The latter represents areas with a minimum energy of the debris flow spreading mostly characterized by muddy sediments, as shown in Figure 8c. The best fit map obtained by the superposition of the real events—recognized in the B8, B10, B11, B12, and B13 watershed basins—and the runout spreading, extracted from the Flow-R modelling processed, allowed us to calibrate the better parameters used in the elaboration of the potential susceptibility map of the Vallo di Diano basin (Figure 12A,B).

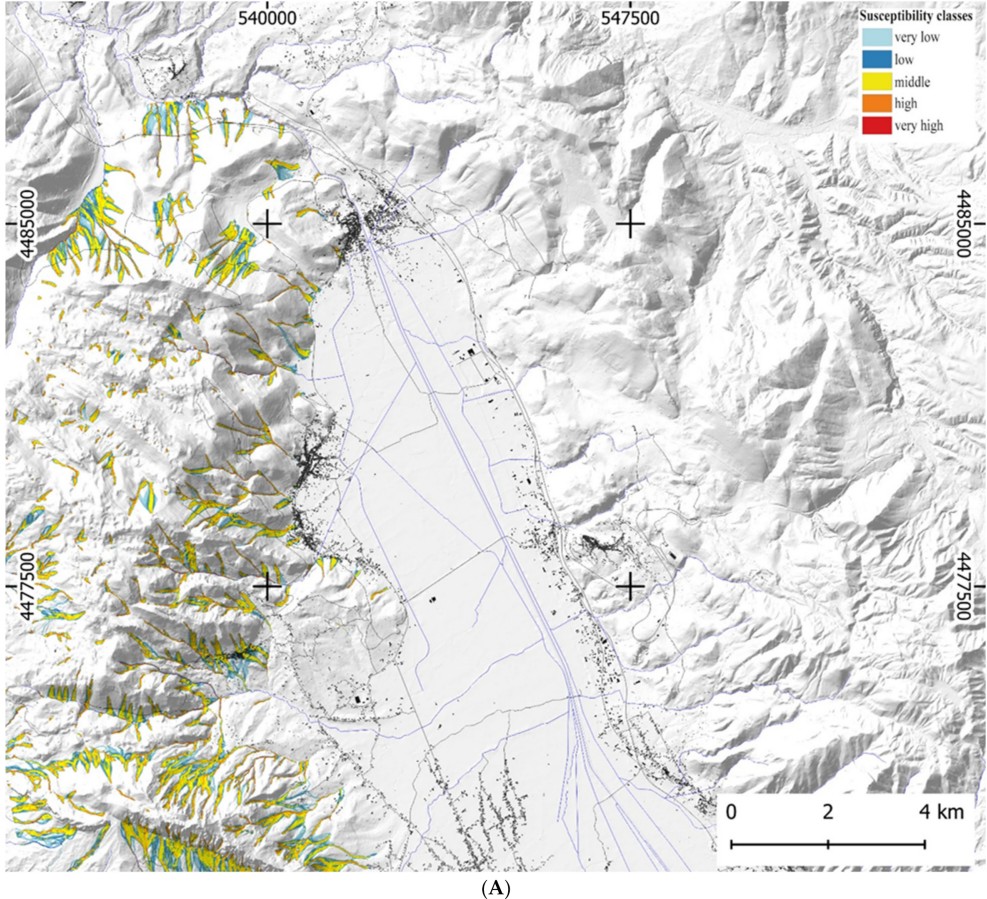

(**A**)

**Figure 12.** *Cont.*

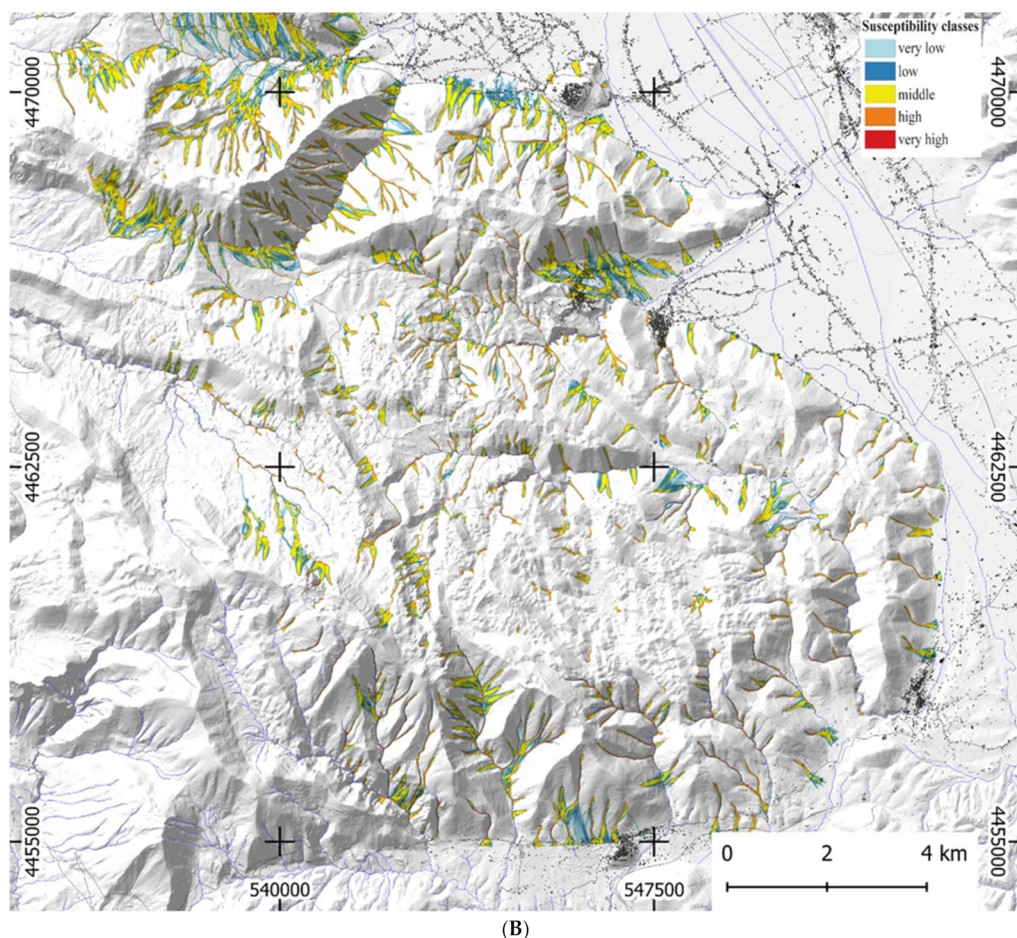

(**B**)

**Figure 12.** Maps of potential susceptibility classes in the western side of the Vallo di Diano basin: (**A**) northern sector, (**B**) southern sector. Dark points are the urban settlements and dark lines are the routes.

## 5. Discussion

A discrimination of debris flows processes rather than debris flood in mountain drainage basins, such as those of the western side of Vallo di Diano, was carried out (Figure 3). In the first approach, the assessment of the basin surface area is important in the discrimination of these two different processes. An empirical relationship has been produced by Heinimann (1998) [39] in Swiss mountain catchments for the lower limit of debris flow source identification and a similar relationship has been proposed by Rickenmann and Zimmermann (1993) [33], who found differences on the boundary of small watershed ranging from 1 to 10 ha. Two different boundaries were found by Horton et al. (2008) [12], the first related to rare events and based on the Heinimann (1988) [39] limit and the second related to extreme events, which is based on the observation of Rickenmann and Zimmermann (1993) [33]. On the western side of the Vallo di Diano basin, the upslope of contributing areas larger than 10 km$^2$ is related to five basins. Two of these catchments, the Marza (B8) and the Buccana (B10) basins have been selected for the investigation because their tributaries were responsible for the 2010–2017 and 1993 debris flows events, respectively. Six catchments with a contributing area ranging from 2.5 and 10 km$^2$ are present, among them the Sinagoga basin (B13) has been investigated because it was affected by the 2005 event. Catchments with a contribution area smaller than 2.5 km$^2$ are eight in the western Vallo di Diano, and two of them, the Valle Torto (B10) and the Valle Cupa (B12) basins, have been selected because they recorded the 1993 debris flows event (Figure 8). Many examples show the discrimination between the debris flood and debris flow component in alluvial fans, plotting the fan slope versus Melton's

index [22,24,26,70]. The identification of debris flow and debris flood processes, comparing the R index in watershed areas and the slope fans has been improved also thanks to other morphometric parameters measured in watershed areas such as the minimum, medium, maximum elevation, and the relief (Table 2). In addition, fan parameters such as area, toe, and concavity also contribute to that discrimination (Table 3).

The Melton index versus fan slope parameters, measured in the 19 watershed basins of the western Vallo di Diano, were directly plotted in the three existing plot diagrams proposed by De Scally and Owens (2004) [24], Wilford et al. (2004) [26], and Santangelo et al. (2006) [27]. The diagram of Santangelo et al. (2006) [27] is the most representative for this study because it is elaborated on similar catchments but placed in the eastern flank of the Vallo di Diano basin. Note that the fast way of identifying a debris flow versus a debris flood based on previous works is not substitutive of field inspection and the analysis of sedimentological and stratigraphical characters of deposits. However, it can be considered as a preliminary approach in large areas where a lot of time is needed for a complete inspection, overall. The plotting of our dataset into the three above mentioned diagrams has revealed that only four basins (B2, B10, B14, and B15 in Figure 4) are in the debris flow process field of all three diagrams (Figure 6a–c). Note that these four basins are smaller than 2.5 km$^2$ and this area value fits well with basins that are most suitable for affecting debris flow processes, as proposed by Horton et al. (2008) [12]. Furthermore, three other basins smaller than 2.5 km$^2$ (B1, B6, and B12 in Figure 4) are included in the debris flow process field of two plots (Figure 6a,c). Overall, seven small basins prone to debris flow have been recognized on the whole western side. It is worth noting that, if we consider the only diagram of Santangelo et al. (2006) [27], then we find sixteen basins prone to debris flows and only three to fluvial floods. Comparing the results furnished by the three plot diagrams and those of the watershed basin areas, we can assign the wider catchments of the B8, B9, B11, B16, and B19 to watersheds mainly prone to debris flood and not to debris flow. The back-analysis applied to the Flow-R modelling was realized on five basins that have recorded the pluviometric events of 1993, 2005, 2010, and 2017. The two catchments are those of the Marza and the Buccana basins (B8 and B11 in Figure 4). As indicated above, they are included within large basins prone to fluvial flood. In this case, the investigation was conducted on the tributaries of the Marza and Buccana main channel, which contain small watersheds smaller than 2.5 km$^2$. These sub-basins are responsible for the activation of debris flow processes that sometimes affect roads and buildings, also producing human deaths. The other three investigated watersheds are the small catchments of the Valle Torto, Valle Cupa, and Sinagoga basins. They are interpreted as basins prone to debris flow processes based on the morphometric analysis. Once the watershed basins prone to debris flows were selected, the corresponding investigation areas were modelled with the Flow-R software, calibrated using the user's maps of the debris flow spreading events that occurred in 1993, 2005, 2010, and 2017 years. The maps have been drawn in GIS to produce an exact overlap with the output map elaborated using the Flow-R modelling. The automatic output maps of the spreading runouts have been realized using the iterative superposition procedure of the two different maps and allowed us to assess the best fit parameters in the five watershed basins. The output raster resulting from the Flow-R modelling has furnished a map of debris flows spreading areas associated with a qualitative probability of potential susceptibility ranging from high to low values. The Flow-R allows us to set the probability of spreading within each cell of the raster from one to zero, where one corresponds to the high potential susceptibility and zero to the lower one [11]. According to Blahut et al. (2010) [13], a classifying representation of the susceptibility maps allows us to identify, in a better way, the areas with a high probability of being reached by debris flows and is useful in the identification of different class values. As suggested by Blahut et al. (2010) [13] we used a classifying representation of the potential susceptibility including five classes from very high to very low. The range values selected in the raster map were reclassified according to a geometrical classification using the QGIS software. The classification included the following range values: (1) 0.0001–0.0025 as very

low; (2) 0.0025–0.0127 as low; (3) 0.0127–0.0556 as middle; (4) 0.0556–0.2366 as high; and (5) 0.2366–1.001 as very high. These values were used in the reclassification of the original susceptibility map obtained using the Flow-R modelling to realize a new map of potential susceptibility on the western side of the Vallo di Diano basin (Figure 12A,B). Further, a computation of the different surface areas, from very low to very high, of the potential susceptibility classes prone to debris flow has been made using the Zonal Istogram tool in the QGis application. The tool allowed us to compute the number of pixels occupying each class of susceptibility in the basins and then, using the 5-meter-resolution DEM analysis, the percentage of the surface area for each class (Figure 13).

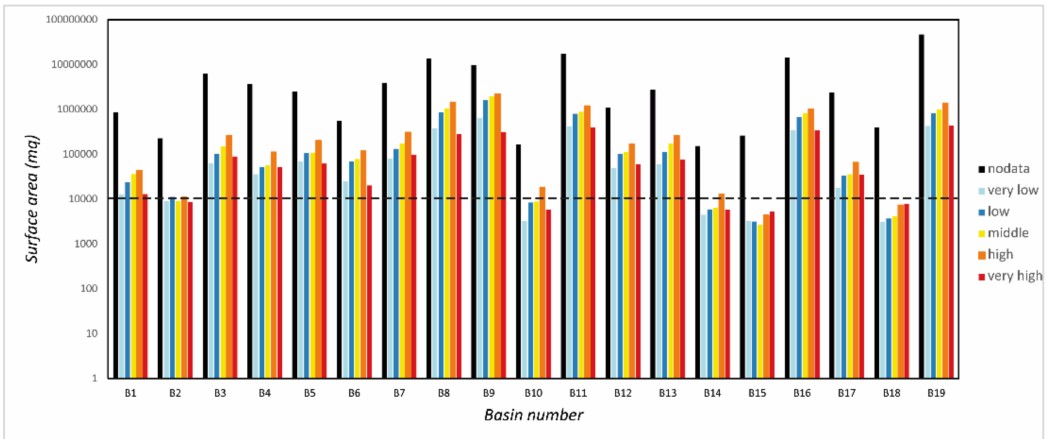

**Figure 13.** Surface areas with different potential susceptibility classes in the 19 watersheds on the western side of the Vallo di Diano basin. Dark dotted line indicates the threshold of 10 km$^2$ in the watersheds. The nodata element in the legend indicate a non-susceptibility class.

The values of the susceptibility classes related to each basin of the studied area reported in the histogram of Figure 13 have shown the extreme variability of value classes in all the basins. A quite similar elaboration, comparing the modelling result and the footprints of the three events, was conducted in the Barcellonette Basin by Kappes et al. (2011) [1], the elaboration provided a 77% match between the affected and modelled areas.

The high susceptibility class values related to the B8, B9, B11, B16, and B19 basins are perfectly coincident with watersheds larger than 10 km$^2$ corresponding to the first category (Figure 4) and could indicate areas with potentially elevated debris flood processes. The analysis of the tributaries' sub-basins in the Marza (B8) and Buccana (B9) have demonstrated that flow processes are mainly due to debris flows. Conversely, the main channel of the B8 and B9 are characterized by fluvial floods and only the upstream left and right-flank sub-basins recorded debris flows. The watershed areas included in the second and third categories—corresponding to the basins included within 2.5 and 10 km$^2$, and <2.5 km$^2$, respectively—are all consistent with high susceptibility classes (Figure 13). The potentially susceptible classes of the five investigated watersheds of the Marza (B8), Valle Torto (B10), Buccana (B11), Valle Cupa (B12), and Sinagoga (B13) basins (Figures 4 and 11) have been analyzed in a pie chart reporting their percentage values (Figure 14).

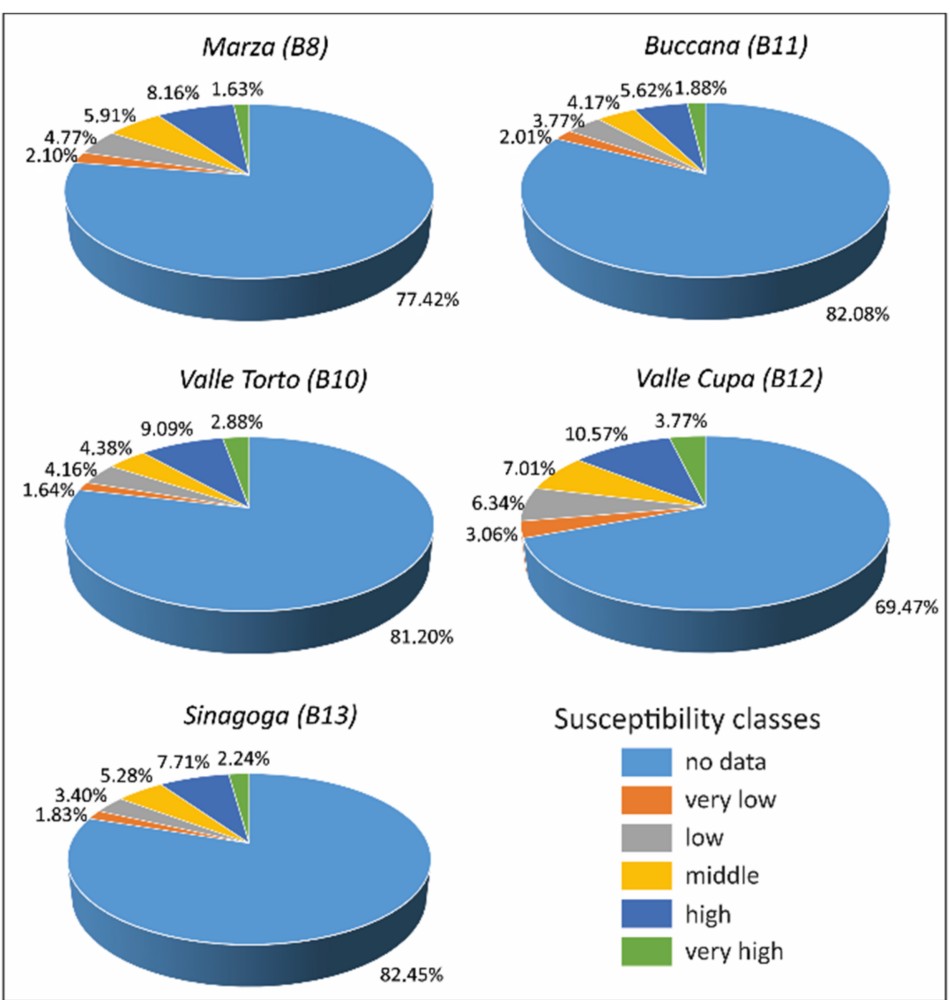

**Figure 14.** Percentage of areas with different potential susceptibility classes in the five collected watersheds.

In the Marza (B8) watershed, the very high susceptibility class covers 1.63% of the whole basin, reaching an area of just 0.28 km$^2$, the high and the middle classes cover 8.16% and 5.91%, respectively, and an area of 2.46 km$^2$, overall. The remaining classes, including the low, the very low, and the no data, represent non susceptibility areas, reaching 84.3% of the whole basin and about 14.75 km$^2$. The Buccana watershed (B11) has revealed a few higher percentages of non-susceptible areas of 87.86%, corresponding to about 18.45 km$^2$. The high classes of susceptibility cover the 11.67% corresponding to 2.44 km$^2$ (Figure 14). Both the B8 and B11 watersheds were included in the first category of basins (>10 km$^2$) mainly related to fluvial processes and have shown similar values of susceptibility classes suggesting the same potential susceptibility. The second category of basins ranging from 2.5 to 10 km$^2$ is related to mixed debris flow/flood processes and is represented by the Sinagoga (B13) watershed. It shows a little increase in the high potential susceptibility classes concerning the previous one reaching a percentage of 16.23% of the total basin area (Figure 14). The non-susceptible areas reach 87.68%, covering an area of 2.89 km$^2$. The third category of basins smaller than 2.5 km$^2$ including the Valle Cupa (B12) and the Valle Torto (B10) watersheds is mainly prone to debris flow processes, as demonstrated by the 1993 event. The Valle Cupa watershed has shown a percentage of 21.35% in the high potential susceptibility classes and an area of 0.34 km$^2$. The classes of non-susceptibility reach 78.87% and an area of 1.25 km$^2$. The high potential susceptibility values in the Valle Torto watershed reach 16.35%, related to an area of 0.03 km$^2$. Conversely, the non-susceptibility classes cover 87% of the whole basin area, corresponding to 0.17 km$^2$ (Figure 14).

Details on the potential susceptibility classes' distribution area in the five basins, also affected by the rainfall events and used in the back-analysis, have revealed that the higher potential susceptibility is located in the Valle Cupa basin, reaching 21.35% and the lower one in the Buccana basin reaching 11.67%. The difference may be due to the fact that debris flow processes in the Buccana basin are exclusively located in the tributary of the flank valley and not in the main channel, whilst the same processes in the Valle Cupa basin are mainly placed in the principal channel and are secondary in the tributaries. Nevertheless, the real distribution of debris flow events in tributaries of the Buccana channel fit quite well with the best simulation elaborated by the model, except in the upstream where the real event was larger than the elaborated map (Figure 15a,b).

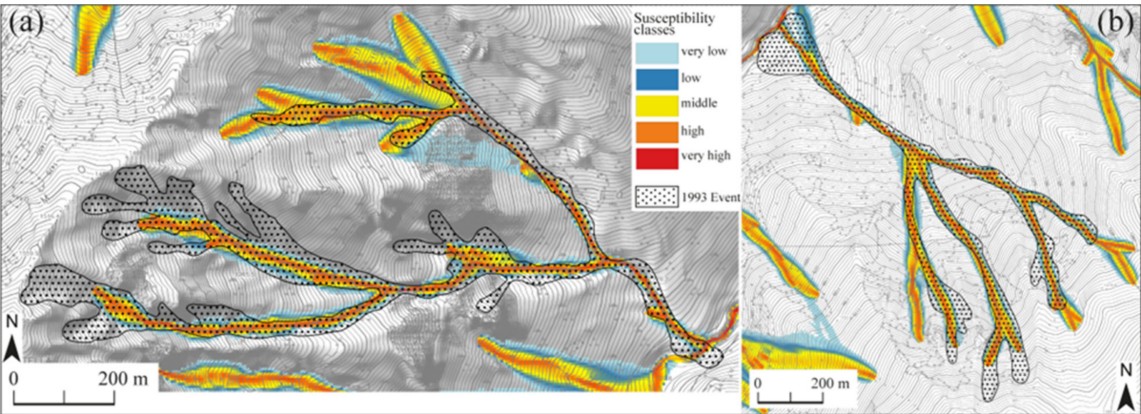

**Figure 15.** Comparison of potential susceptibility classes and the 1993 debris flow mapped event in the left (**a**) and right (**b**) tributaries sub-basins of the Buccana watershed.

In the Valle Cupa watershed, the comparison of the event of 1993 and the modelling of debris flow spreading has revealed a very similar result showing a similar distribution of runout (Figure 16a). Note the threshold in the runout indicated by middle to high susceptibility classes in correspondence with the road S.P. n. 11 and related buildings that hinder the distribution of flow, thus decreasing the susceptibility class (Figure 16b).

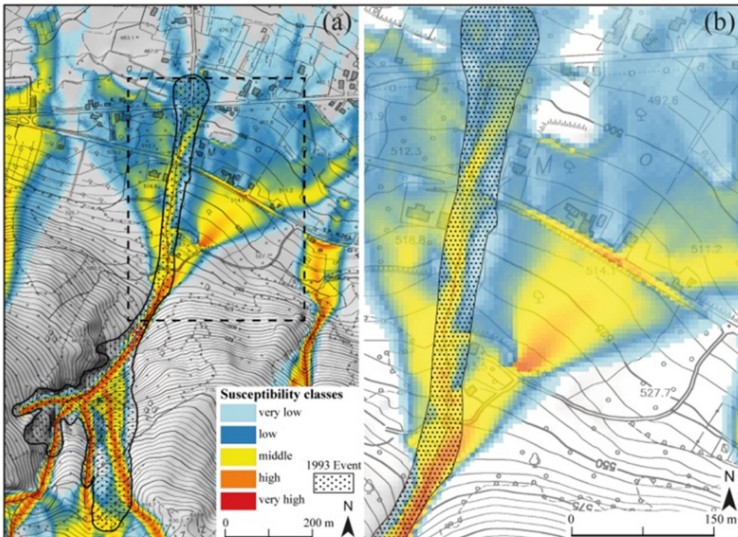

**Figure 16.** Comparison of potential susceptibility classes and the 1993 debris flow mapped event in the Valle Cupa watershed (**a**). Details of the debris flow spreading in the fan are shown in (**b**).

In the sub-basins of the Marza watershed, the superposition has shown a best fitting; the real event of the 2017, which is mainly channelized, is well recognized by the Flow-

R modelling and particularly the very high classes of potential susceptibility map are corresponding to the real depositional debris flow event (Figure 17a–d).

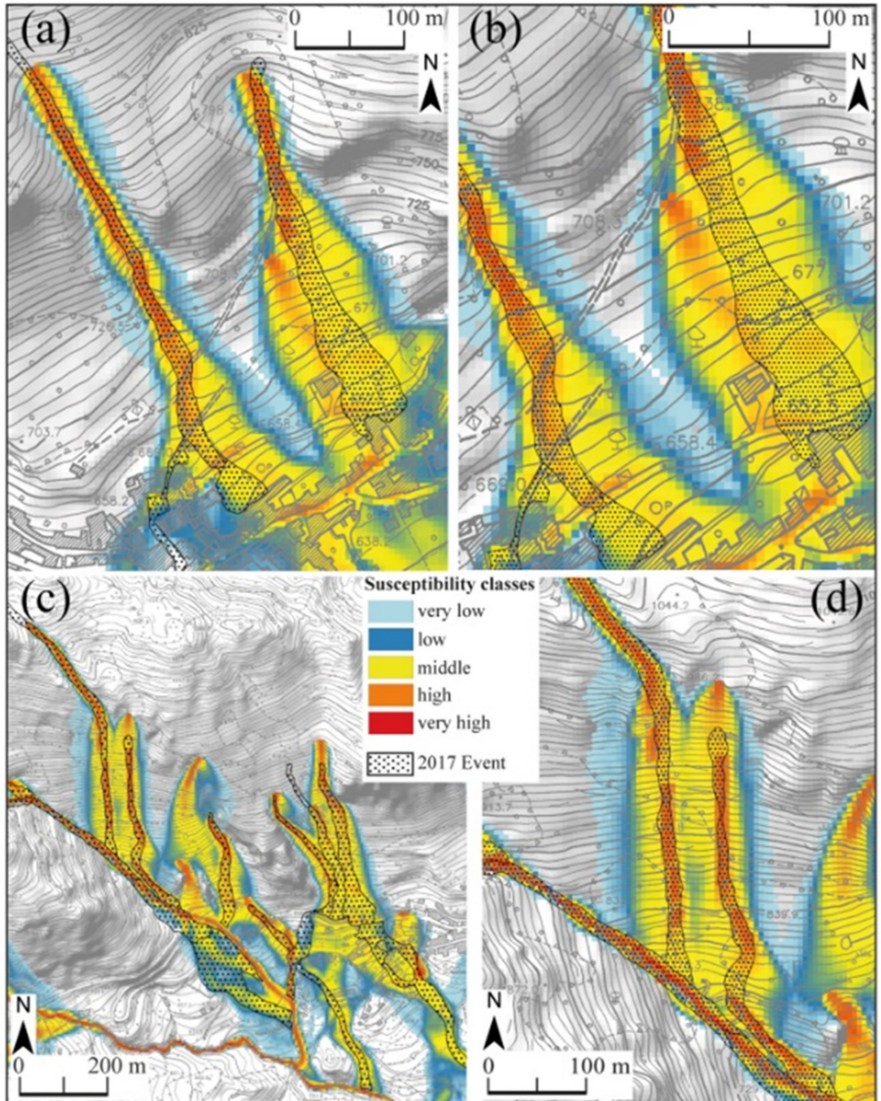

**Figure 17.** Comparison of potential susceptibility classes and the 2017 debris flow mapped event in the Marza sub-basins (**a**,**c**). Details are reported in (**b**,**d**), respectively.

## 6. Conclusions

The morphometric analysis of the watershed basins (Figure 4) in the study area provided preliminary information about the dominant process of debris flows or debris floods characterizing each basin. The first element of discrimination is represented by the basin area, which allowed many authors to identify the debris flows rather than the debris floods processes. The susceptibility mapping modelled with the Flow-R software compared with the mapped flooding events that occurred on several years in the selected watershed basins has shown a quite good correspondence, proving the reliability of the automatic elaboration and map output. In this case, the mapping could significantly contribute to the evaluation in the first approximation of dangerous areas. However, it is certain that the field knowledge of the studied areas is essential for the assessment of susceptibility maps and also for the evaluation of correct automatic modelling.

On the contrary, the efficiency of the automatic modelling results become less relevant. The Flow-R output map has demonstrated a good approximation of high susceptibility classes and the real debris flow events that occurred in relatively small areas of the Vallo di



Diano. Take into account that the triggering conditions of debris flow events are highly variable and, therefore, some singular condition needs to be better investigated in the field. This means that for the correct modelling of runout analysis, a very precise bound delimitation by an expert of the areas' susceptibility to debris flow is necessary.

In conclusion, the automatic mapping of an areas' susceptibility to debris flow with the Flow-R modelling has proven to be quite reliable and can be considered the first and preliminary step in the analysis of large areas affected by debris flow processes. Indeed, it provides the recognition of areas that need to be investigated with more detail.

The evaluation of the spreading parameters controlling the debris flow, the back-analysis of the propagation events realized through an iterative runs of the Flow-R model, and the computation of morphometric parameters allowed us to test many setting values with the real source areas and propagation events. The best-fit simulation realized for five sampled watersheds, really affected by pluviometric events, has demonstrated the good reliability of the Flow-R modelling in a preliminary analysis of the debris flow processes affecting large areas. Note that good results of the method are certainly linked to the accuracy of the field survey by users and to the exact elaboration of the GIS software.

**Author Contributions:** Conceptualization, S.I.G., E.P. and V.S.; methodology, S.I.G., E.P. and V.S.; software, S.I.G. and V.S.; formal analysis, S.I.G., E.P. and V.S.; investigation, S.I.G., E.P. and V.S.; data curation, S.I.G., E.P. and V.S.; writing—original draft preparation, S.I.G. and E.P.; writing—review and editing. All authors have read and agreed to the published version of the manuscript.

**Funding:** This research was funded by FFABR 2018 granted to Salvatore Ivo Giano and by "Eni-Val d'Agri Project" founded to Giacomo Prosser.

**Institutional Review Board Statement:** Not applicable.

**Informed Consent Statement:** Not applicable.

**Conflicts of Interest:** The authors declare no conflict of interest.

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
