# Peer review of "Morphometry and Debris-Flow Susceptibility Map in Mountain Drainage Basins of the Vallo di Diano, Southern Italy"

_remotesensing, doi:10.3390/rs13163254_

Round 1

Reviewer 2 Report

Dear authors,

In my opinion, the subject of this work is relevant for the Remote Sensing journal readers, and sufficiently novel and interesting to warrant publication. The research questions are very important in various fields of scientific and applied investigation. The work sound and the overall approach envisioned and implemented are generally correct. The analyses and discussion are the logical outcome of the presented data.

All the key-elements (i.e., abstract, introduction, methodology, results, discussion, and conclusions) are present, however it is sometimes difficult to follow the authors' reasoning in organizing the text. The manuscript often assumes that the reader is more familiar with the evidence presented than is realistic to expect. It is my opinion that the strategy and study design must be much better explained and structured. The paper has a good emphasis on explaining how the flow-R model work and too little on describing practically how it was implemented. The authors need more specific detail on the implementation of the model (e.g., lines 250-259). These things are certainly clear and obvious in the minds of the authors but need to be explained more in the methodological section. Maybe, a flow diagram (in a graphical format), showing a synthesis of all the methodological steps, would be of great help to the reader. Moreover, the authors need to include a table with information regarding typology, scale (or resolution), classes and source of the collected spatial dataset used to implement the models, as well as a resumable table of the used morphometric index.

At some points, the manuscript is confused and not very well organized with a mix of results-discussion-methods (i.e., in a clear and concise results section no bibliographic references should be necessary). Indeed, in the results section the following texts are discussion or methods: lines 388-394, 418-427, 430-437, 441-450, 483-484, 547-559, 565-571, 577-606 (see also the place of previous lines 51-55).

Figures and tables are all necessary but with a graphical aspect which prevent a correct reading. The authors need to dedicate more attention to the graphical and cartographic expressions of the maps: Fig. 2 need the hillshade in transparency, like Fig. 3; Figs. 1, 14, 15 and 16 are too small and do not read well; in Figs. 1 and 11 is missing to indicate the north direction.

  • See Line 100: change Bf with Br.
  • See Line 167: explain why 10m-resolution if the DEM was constructed with 5-m resolution (see line 85).
  • Considering that the discussion section is very long, it would be better to separate discussion and Final remarks (eventually, also to add “management implications” section).

I recommend at the authors to seek English science editor to assist in the revision to improve the style, grammar and restructuring of sentences and paragraphs (minor spell check required) to make the manuscript more precise and clearer for readers.

In summary, I think that this manuscript could be accepted as an article in the Remote Sensing journal only after some revisions.

Reviewer 3 Report

Dear Authors,

The article Morphometry and debris-flow susceptibility map in mountain drainage basins of the Vallo di Diano intermontane basin, Southern Italy (ID: remotesensing-1337224) presents the results of a statistical method to identify areas susceptible to debris-flow in 19 mountain catchments in southern Italy. At this point, the article needs major revisions/rejection. This depends on the editor's decision. Because, as it is stated in the abstract, the paper presents a methodology that was already developed for this purpose. My opinion is that the paper does not bring anything novel and does not qualify itself to be published in Remote Sensing (unless it was submitted for a Special Issue).

In the Introduction section, you should offer a straightforward definition of debris flow and debris flood, with the appropriate references.

L12: this sentence sounds odd; try rephrasing

L86: what does I.G.M. stand for?

L94: it would look better if you put these morphometric parameters in a table

You should move section 3 (Geological and Geomorphological settings) before section 2 (Materials and Methods)

Figure 1: can be enlarged for better clarity

Figure 3: improve the resolution of the image and include coordinates

Figure 6: here you overlap your results over some old maps (however, this is not mentioned in the methodology section); please, correct

Figure 11: the map does not have a scale. Then, you should include at least the anthropogenic factors (human settlements, roads) that debris flow might affect. Like it is, the map has no applicability or importance in the real world or for local authorities (which is a thing that you should add towards the end of your paper).

References are not formatted according to the RS instructions for authors; please, correct them.

Good luck with the review!

Reviewer 4 Report

See attached document for detailed comments. In general:

Explain the difference between debris flow, debris flood and fluvial flood. Add a scheme or figure. Explain physical meaning of Melton ratio.

Some descriptions in this article are much too detailed for general interest of readers such as geology, local geomorphic parameters, specific software parameters, please reduce text and avoid repeating.

How can we interpretate these results in terms of risk or damage probability? Meteorological info has to be added. Or is the risk of high rainfall events evenly distributed? Please comment.

Review English language, add comma's and cut lines to improve clearness

Round 2

Reviewer 1 Report

The MS is improved compared to the original one.
Though I appreciated the revision efforts made by the Authors. hence I accept it in the present form.

Reviewer 3 Report

Dear Authors,

You have improved your manuscript according to the reviewer's recommendations. Now it looks and sounds much better. Therefore, I think it can be forwarded to the next step of the publication process.

Kind regards.